# Global dataset of thermohaline staircases obtained from Argo floats and Ice-Tethered Profilers

Carine G. van der Boog[1], J. Otto Koetsier[1], Henk A. Dijkstra[2], Julie D. Pietrzak[1], and Caroline A. Katsman[1]

[1]Environmental Fluid Mechanics, Civil Engineering and Geosciences, Delft University of Technology, Delft, the Netherlands
[2]Institute for Marine and Atmospheric research Utrecht, Utrecht University, Utrecht, the Netherlands

**Correspondence:** Carine van der Boog (c.g.vanderboog@tudelft.nl)

**Abstract.** Thermohaline staircases are associated with double-diffusive mixing. They are characterised by stepped structures consisting of mixed layers of typically tens of meters thick that are separated by much thinner interfaces. Through these interfaces enhanced diapycnal salt and heat transport take place. In this study, we present a global dataset of thermohaline staircases derived from observations of Argo profiling floats and Ice-Tethered Profilers using a novel detection algorithm. To establish the presence of thermohaline staircases, the algorithm detects subsurface mixed layers and analyses the interfaces in between. Of each detected staircase, the conservative temperature, absolute salinity, depth and height, as well as some other properties of the mixed layers and interfaces are computed. The algorithm is applied to 487,493 quality-controlled temperature and salinity profiles to obtain a global dataset. The performance of the algorithm is verified through an analysis of independent regional observations. The algorithm and global dataset are available at https://doi.org/10.5281/zenodo.4286170.

## 1 Introduction

Thermohaline staircases consist of subsurface mixed layers that are separated by thin interfaces. They are associated with double-diffusive processes, which in turn result from a two orders of magnitude difference between the molecular diffusivity of heat and that of salt (Stern, 1960). Whenever the vertical gradients of temperature- and salinity-induced stratification have the same sign, these differences in molecular diffusivity can enhance the vertical mixing through double-diffusive convection, leading to effective diffusivities of the order of $10^{-4}$ m$^{-2}$ s$^{-1}$ (Radko, 2013, and references therein).

It is still a topic of discussion how double-diffusive convection leads to the formation of thermohaline staircases in oceanic environments (Merryfield, 2000). For example, Stern (1969) argued that small-scale mixing processes trigger the formation of internal waves. On the other hand, variations in the turbulent heat and salt fluxes (Radko, 2003) or in the counter-gradient buoyancy fluxes that sharpen density gradients (Schmitt, 1994) could also lead to the formation of thermohaline staircases. Lastly, subsurface mixed layers can also arise from thermohaline intrusions (Merryfield, 2000). Although it remains unclear

how these staircases arise, these studies agree that the formation of these subsurface mixed layers are related to double-diffusive processes.

Based on the Turner angle ($Tu$), which compares the density component of the temperature distribution with the density component of the salinity distribution, two regimes of double diffusion can be distinguished (Ruddick, 1983). Waters with $-90° < Tu < -45°$ correspond to a stratification where both temperature and salinity increase with depth and belong to the diffusive-convective regime (DC). Those with $45° < Tu < 90°$ correspond to a stratification where temperature and salinity decrease with depth and belong to the salt-finger regime (SF).

Theoretical and laboratory studies have indicated that diapycnal fluxes of heat and salt in thermohaline staircases are elevated compared to the background turbulence (e.g., Schmitt, 1981; Kelley, 1990; Radko and Smith, 2012; Garaud, 2018). These results were confirmed by a tracer release experiment in the western tropical Atlantic Ocean (Schmitt, 2005). Although these enhanced fluxes were observed, the importance of these fluxes for the global mechanical energy budget remain unknown. Moreover, the vertical heat and salt fluxes in thermohaline staircases can also affect water-mass properties. In some regions, persistent thermohaline staircases with layers stretching over a few hundred kilometers have been observed (Schmitt et al., 1987; Timmermans et al., 2008; Shibley et al., 2017), which could result in significant diapycnal fluxes between water masses. For example, the double-diffusive diapycnal fluxes in the Mediterranean Sea dominate the transport between the deep water masses (Zodiatis and Gasparini, 1996; Bryden et al., 2014; Schroeder et al., 2016), and in the Arctic Ocean and Southern Ocean, an upward heat flux has been observed through staircase interfaces (Timmermans et al., 2008; Shibley et al., 2017; Polyakov et al., 2012; Bebieva and Speer, 2019).

Modelling studies that incorporated parameterizations of double-diffusive fluxes, indicated that the associated double-diffusive diapycnal fluxes can reduce the strength of the global overturning circulation (Gargett and Holloway, 1992; Merryfield et al., 1999; Oschlies et al., 2003). To be able to study this with observations, we present a global dataset of the occurrence of thermohaline staircases and their properties. The dataset is based on observations from Argo floats and Ice-Tethered Profilers. In the following sections we briefly describe the raw data used to extract the dataset (Section 2) and the algorithm we designed to detect staircase structures (Section 3). The sensitivity of this detection algorithm to the chosen input parameters is assessed in Section 4. The dataset is verified in Section 5, followed by some guidelines for the use of the dataset in Section 6.

## 2 Data preparation

The dataset contains observations of autonomous Argo floats and autonomous Ice-Tethered Profilers (ITP). The data of all active and inactive profilers is obtained from http://www.argo.ucsd.edu and http://www.whoi.edu/itp from 13 November 2001 to 14 May 2020. Details on the profilers are described in Krishfield et al. (2008) and Toole et al. (2011) for the ITP and in Argo (2020) for the Argo floats. First a quality check is performed, where a profile is excluded from analysis if it was taken by an Argo float mentioned on the grey list. This grey list contains floats that may have problems with at least one of the sensors (https://www.nodc.noaa.gov/argo/grey_floats.htm). As thermohaline staircases consist of mixed layers with depths of tens of meters, we also require that profiles have continuous data up to 500 dbar with an average resolution finer than 5 dbar. Details

**Table 1.** Number of floats and profiles in the global dataset. Profiles taken with Argo floats are categorised by the Data Assembly Center (DAC). Profiles taken with Ice-Tethered Profilers are categorised as ITP. The percentage between brackets indicates the relative contribution to the total number of profiles in the global dataset (487,493 profiles). More details on abbreviations of DAC can be found in Argo (2019)

| DAC / ITP | number of floats | profiles |
|-----------|------------------|----------|
| aoml      | 2,692            | 312,285 (64.1 %) |
| bodc      | 93               | 11,092 (2.3 %) |
| coriolis  | 347              | 27,134 (5.6 %) |
| csio      | 81               | 15,099 (3.1 %) |
| csiro     | 378              | 42,942 (8.8 %) |
| incois    | 65               | 4,363 (0.9 %) |
| jma       | 205              | 22,919 (4.7 %) |
| kma       | 1                | 1 (0.0 %) |
| kordi     | 0                | 0 (0.0 %) |
| meds      | 145              | 9,285 (1.9 %) |
| nmdis     | 0                | 0 (0.0 %) |
| ITP       | 82               | 42,373 (8.7 %) |

on the origin and vertical resolution of the profiles are depicted in Table 1 and Figure 1, in which Figure 1b confirms that all profiles have observations deeper than 500 dbar. Furthermore, the average vertical resolution of the profiles indicates the average resolution is well below the 5 dbar that was used as a threshold (Fig. 1c). After this quality control, 487,493 vertical temperature and salinity profiles remain. Their global distribution is shown in Figure 2.

Next, the profiles of the Argo floats and ITP were linearly interpolated to a vertical resolution of 1 dbar from the surface to 2000 dbar so that their data could be analysed in a consistent manner. As a result, the small steps in, for example, Arctic staircases might be missed (see Section 5). From these interpolated profiles we calculate several variables. Absolute salinity (S) in g kg$^{-1}$ and conservative temperature (T) in °C are computed with the TEOS-10 software (McDougall and Barker, 2011). Note that we use conservative temperature as this is more accurate than potential temperature in computations concerning heat fluxes and heat content (Graham and McDougall, 2013). We apply a moving average of 200 dbar (Table 2) to obtain the background conservative temperature and absolute salinity profiles of the water column and to compute the thermal expansion coefficient ($\alpha$ in °C$^{-1}$) and the haline contraction coefficient ($\beta$ in kg g$^{-1}$). A consequence of the moving average of 200 dbar is that the upper 100 dbar and lower 100 dbar of each profile is omitted in the remainder of the analysis. The Turner angle is computed using profiles that were smoothed with a moving average of 50 dbar instead of 200 dbar, which is similar to Shibley et al. (2017), following Ruddick (1983), from

$$Tu = \tan^{-1}\left(\alpha\frac{\partial T}{\partial p} - \beta\frac{\partial S}{\partial p}, \alpha\frac{\partial T}{\partial p} + \beta\frac{\partial S}{\partial p}\right), \tag{1}$$

where the vertical gradients are approximated with a central differences scheme.

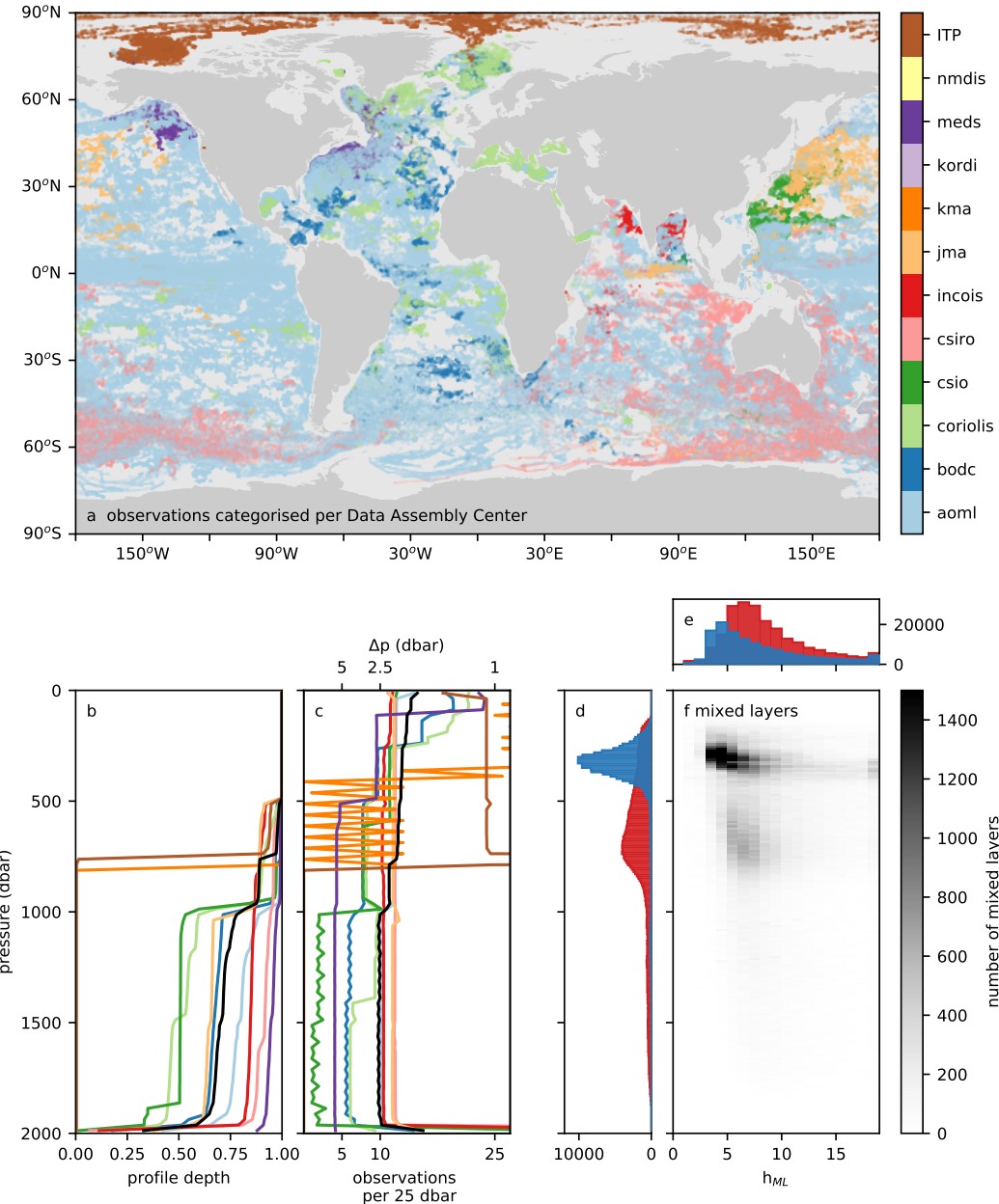

**Figure 1.** (a) Locations of observations categorised by Data Assembly Centers (DAC) when obtained by an Argo float. Profiles obtained with Ice-Tethered Profilers are indicated with ITP. (b) Cumulative fraction of profiles that reached a given pressure in 25-dbar intervals from 0 to 2,000 dbar per DAC. (c) Average number of observations in 25-dbar intervals from 0 to 2,000 dbar. (d) Distribution of detected mixed layer pressures in the salt-finger (red histogram) or diffusive-convective (blue histogram) regime. (e) Number of detected mixed layers height in the salt-finger (red histogram) or diffusive-convective (blue histogram) regime. (f) Distribution of detected mixed layer heights in thermohaline staircases per pressure level. Panels (b) and (c) were obtained following Wong et al. (2020). Black lines indicate the averages in total global dataset. More details on abbreviations of DAC can be found in Argo (2019)

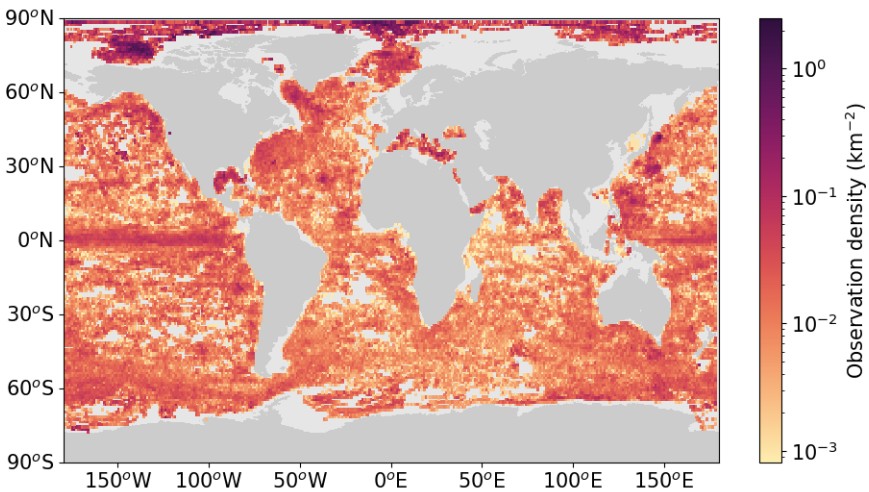

**Figure 2.** Observation density of the number of profiles obtained from the Argo floats and Ice-Tethered Profilers after quality control (in km$^{-2}$). Observation density is binned per degree longitude and degree latitude. Empty bins indicate that no data was available at that location.

**Table 2.** Input parameters applied during the data preparation and the algorithm as used in this study. The sensitivity of the output of the algorithm to the input variables is discussed in the Section 4.

| parameter | description | value |
|---|---|---|
| moving average window | chosen to obtain background profiles | 200 dbar |
| $\partial\sigma_1/\partial p_{max}$ | density gradient threshold for detection mixed layer | 0.0005 kg m$^{-3}$ dbar$^{-1}$ |
| $\Delta\sigma_{1,ML,max}$ | maximum density gradient within mixed layer | 0.005 kg m$^{-3}$ |
| $h_{IF,max}$ | maximum interface height | 30 dbar |

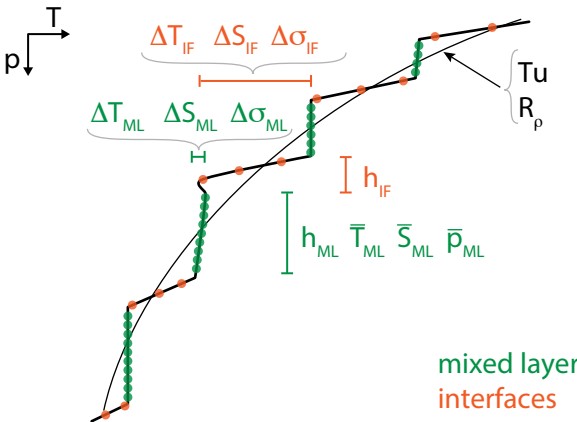

**Figure 3.** Schematic of a typical temperature profile with staircases, indicating the definitions of the quantities used to detect the thermohaline staircases (green: mixed layer; orange: interface).

## 3 Detection algorithm

After the data pre-processing, we apply a detection algorithm that exploits the vertical structure of staircase profiles (Fig. 3). The benefit of using the vertical structure, instead of using assumptions based on the Turner angle, is that we can use this angle to verify the results. The detection algorithm consists of five steps. First the algorithm detects all data points that are located in the subsurface mixed layers (ML, green dots in Fig. 3) by identifying weak vertical density gradients in conservative temperature and absolute salinity. Next, the properties of any layer lying between the mixed layers (the interfaces, IF, orange dots in Fig. 3) are assessed by applying a minimum in temperature and salinity variations. Third, the height of the interface and variations within the interface are limited. The fourth step determines the regime of double diffusion (diffusive convection or salt fingers), and the fifth step is the identification of sequences of interfaces, which eventually characterises the thermohaline staircases. The different steps of the algorithm applied to three example profiles are shown in Figures A1-A3. In the following subsections, each algorithm step is described in more detail.

### 3.1 Mixed layers

The first step of the detection algorithm is the identification of the mixed layers. Preferably, this is done by assessing a density difference relative to a reference pressure, which is the most reliable method to detect a mixed layer (Holte et al., 2017). However, in the case of a thermohaline staircase it is necessary to detect subsurface mixed layers, because the reference pressure is unknown beforehand. To determine this reference pressure, a threshold gradient criterium is applied first (Dong et al., 2008). In this criterium, vertical density gradients are identified as a mixed layer whenever the gradients are below a certain threshold.

We apply the gradient criterium on the vertical gradients of the potential density anomaly at a reference pressure of 1000 dbar ($\sigma_1$). We used a threshold of $\partial \sigma_1 / \partial p_{max}$ =0.0005 kg m$^{-3}$ dbar$^{-1}$ (Table 2), which is similar to mixed layer gradients used by Bryden et al. (2014). Furthermore, this threshold gradient is slightly larger than the threshold used by Timmermans et al. (2008), who used 0.005°C m$^{-1}$ (which corresponds to $\partial \sigma_1 / \partial p_{max}$ =0.00036 kg m$^{-3}$ dbar$^{-1}$). The threshold gradient method is applied on both conservative temperature and absolute salinity profiles, i.e.,

$$\left| \alpha \rho_0 \frac{\partial T}{\partial p} \right| \leq 0.0005 \text{ kg m}^{-3}\text{dbar}^{-1},$$

$$\left| \beta \rho_0 \frac{\partial S}{\partial p} \right| \leq 0.0005 \text{ kg m}^{-3}\text{dbar}^{-1}. \tag{2}$$

Also the vertical density gradients from the combined temperature and salinity effects must satisfy this condition:

$$\left| \frac{\partial \sigma_1}{\partial p} \right| \leq 0.0005 \text{ kg m}^{-3}\text{dbar}^{-1}. \tag{3}$$

These three conditions ensure that the vertical conservative temperature, absolute salinity and potential density gradients are all below the threshold value. At each pressure level where all three conditions are met the datapoint is identified as a mixed
layer. Next, for each continuous sequence of data points, the algorithm computes the average pressure. This is then used as a reference pressure, which is required to be able to apply the mixed layer detection.

    At every reference pressure, a maximum density range is required within the mixed layers to identify the full vertical extent of each mixed layer. To allow for small variations of conservative temperature and absolute salinity in the mixed layer, but to exclude variations in the interface, we use a threshold of $\Delta\sigma_{1,ML,max}$ =0.005 kg m$^{-3}$ for density variations within each
mixed layer (Table 2). This density range corresponds to the density range used by Holte et al. (2017) for the detection of surface mixed layers. The applied density range allows for mixed layers with heights of the order of 10 m assuming gradients of $\partial \sigma_1 / \partial p_{max}$ =0.0005 kg m$^{-3}$ dbar$^{-1}$. To ensure separation between individual mixed layers, the upper and lower datapoint of each mixed layer are removed. Note that this results in a minimum interface height of 2 dbar, which could result in false negatives in for example the Arctic Ocean (Section 5)

After applying the threshold for density range, the algorithm defines each continuous set of datapoints as a mixed layer and computes the average pressure ($\overline{p}_{ML}$), average conservative temperature ($\overline{T}_{ML}$), average absolute salinity ($\overline{S}_{ML}$), mixed layer density ratio ($\overline{R}_\rho = \alpha \frac{\partial \overline{T}}{\partial p} / \left( \beta \frac{\partial \overline{S}}{\partial p} \right)$), average Turner angle ($\overline{Tu}_{ML}$) and height ($h_{ML}$) for each mixed layer.

### 3.2   Interfaces: conservative temperature and absolute salinity variations

The algorithm defines an interface as the part of the water column between two mixed layers. In addition, to ensure a stepped
structure the algorithm requires that the conservative temperature, absolute salinity and potential density variations within each mixed layer should be smaller than the variations in the interface (Fig. 3):

$$\max\left( |\Delta T_{ML,1}|, |\Delta T_{ML,2}| \right) < |\Delta T_{IF}|;$$

$$\max\left( |\Delta S_{ML,1}|, |\Delta S_{ML,2}| \right) < |\Delta S_{IF}|;$$

$$\max\left( |\Delta \sigma_{1,ML,1}|, |\Delta \sigma_{1,ML,2}| \right) < |\Delta \sigma_{1,IF}|; \tag{4}$$

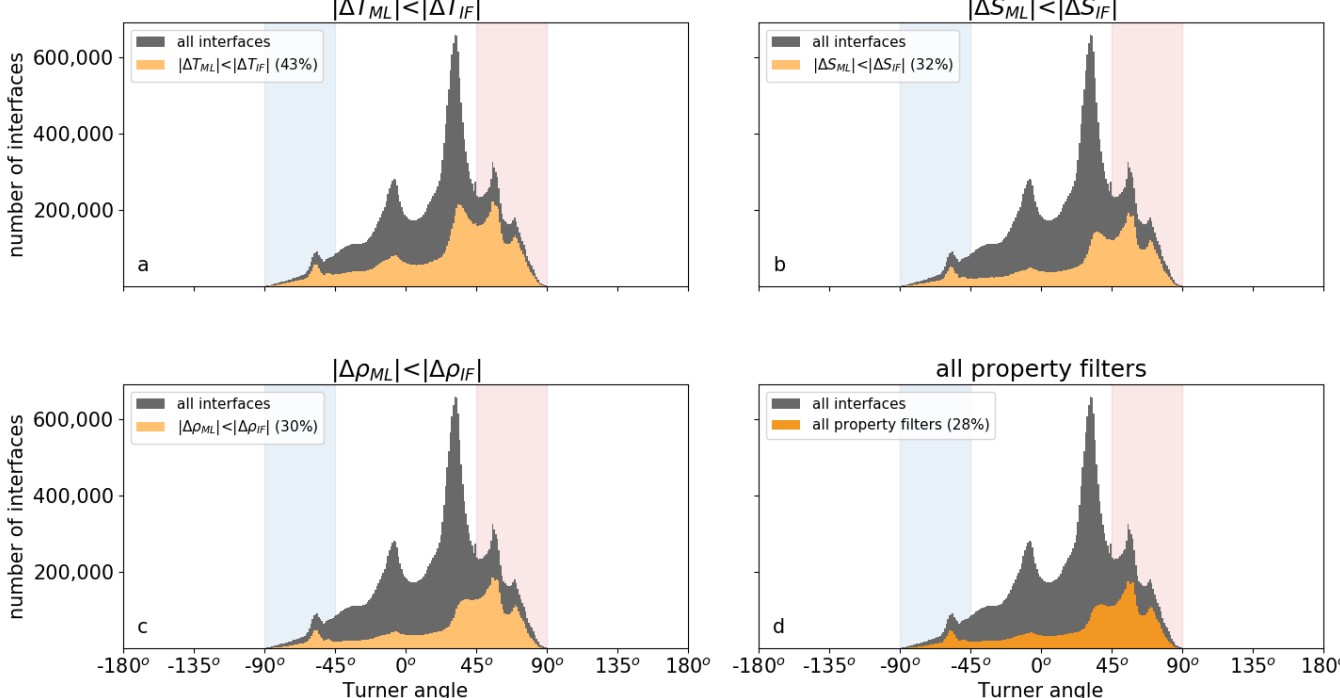

**Figure 4.** Histogram of the number of detected interfaces as a function of the Turner angle (Tu) by applying a criteria for (a) conservative temperature, (b) absolute salinity, (c) potential density and (d) all three properties given in equation 4 (orange shading). Each panel shows the data remaining compared to the raw interface data (grey). Vertical shaded bands correspond to Turner angles in the diffusive-convective (blue) and salt-finger (red) regime.

where the subscripts 1 and 2 correspond to the mixed layer directly above and below an interface, respectively. It appears that most data points that meet these requirements (orange histograms in Fig. 4a-c) have Turner angles in the two double-diffusive regimes. This dependence of the variations in the interfaces on the Turner angle is in line with expectations that staircase-like structures are mostly found within double-diffusive regimes. In total, 28 % of all detected interfaces meet all three requirements (Fig. 4d).

### 3.3 Interface: height

The next step in the staircase detection algorithm is to limit the height of the interface to ensure that the mixed layers are separated from each other by a relatively thin interface (Fig. 3). We require

$$h_{IF} < \min\left(h_{ML,1}, h_{ML,2}\right), \tag{5}$$

i.e. the interface height is smaller than the height of the mixed layers directly above and below the interface. In total, 27 % of the interfaces that fulfilled the conservative temperature and absolute salinity requirements meet this requirement (Fig. 5a).

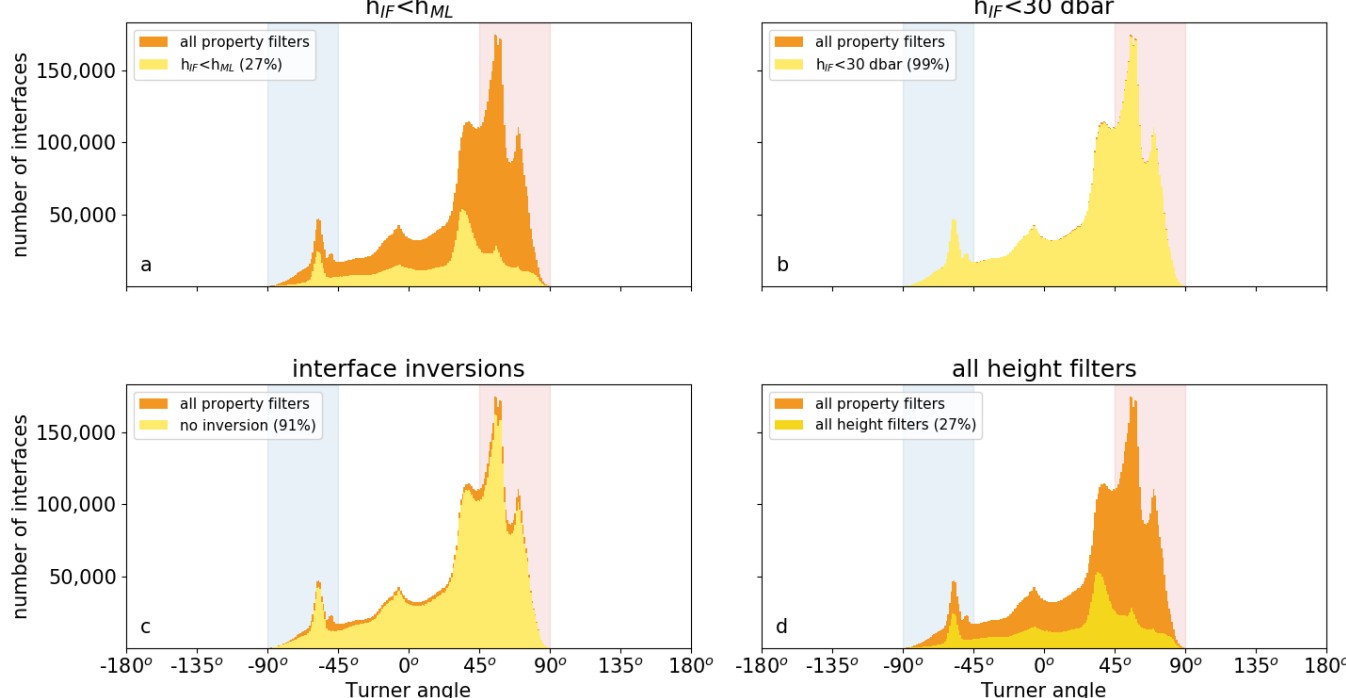

**Figure 5.** Histogram of the number of detected interfaces as a function of the Turner angle (Tu) by applying a criteria for (a) height, (b) maximum height, (c) inversions and (d) all three height limitations (yellow shading). Each panel shows the data remaining compared to the interfaces detected based on the conservative temperature and absolute salinity requirements shown in Fig. 4d (orange shading). Vertical shaded bands correspond to Turner angles in the diffusive-convective (blue) and salt-finger (red) regime.

Note that this part of the algorithm defines the top and bottom of a sequence of a staircase in a profile. Furthermore, the tallest observed interfaces are found in the Mediterranean Sea with heights up to $h_{IF} = 27$ m, where they separate mixed layers of over 100 m (Zodiatis and Gasparini, 1996; Radko, 2013). To prevent false detection of large vertical interfaces of up to hundreds of meters, we limit the interface height to $h_{IF,max} = 30$ dbar (Table 2, Fig. 5b). This only affects the classification of 1 % of the interfaces (Fig. 5b).

To solely detect step-like structures that are associated with the presence of thermohaline staircases, the algorithm also removes all interfaces with conservative temperature or absolute salinity inversions. This is done by limiting the number of local minima and maxima of the conservative temperature and absolute salinity allowed in each interface to two (Fig. 5c). The combination of all three interface height requirements is met by 27 % of the interfaces detected based on the conservative temperature and absolute salinity requirements discussed in the previous section (Fig. 5d).

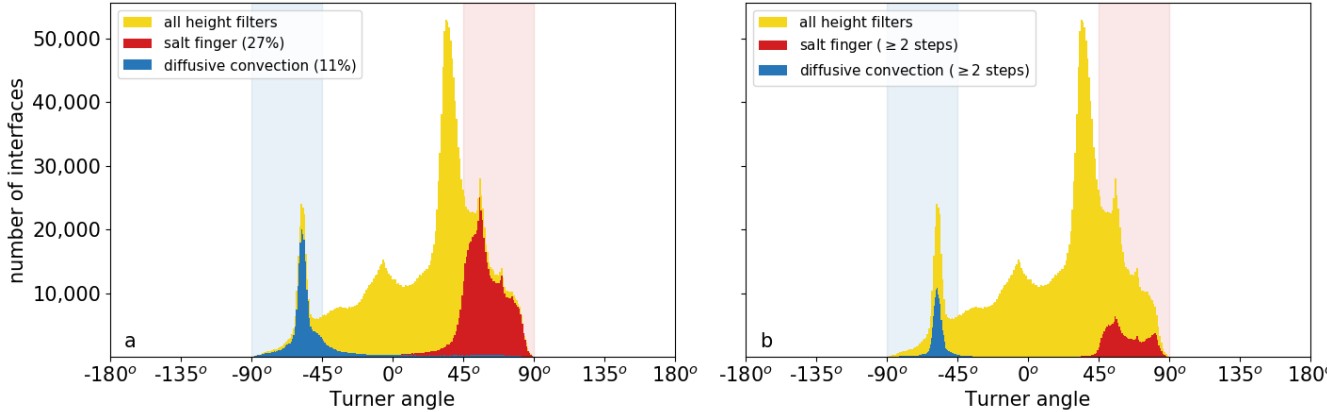

**Figure 6.** Histogram of the number of detected interfaces as a function of the Turner angle (Tu) after (a) classification of the double-diffusive regime and (b) selection of sequences of the interfaces. Each panel shows the data remaining compared to the interfaces detected based interface height requirement shown in Fig. 5d (yellow shading). Vertical shaded bands correspond to Turner angles in the diffusive-convective (blue) and salt-finger (red) regime.

### 3.4 Interface: double-diffusive regime

After the algorithm has selected all the interfaces with a step-like structure, the double-diffusive regime of each interface is assessed (Fig. 6a). In case both conservative temperature and absolute salinity of the mixed layers above and below the interface increase with pressure, the interface is classified as the diffusive-convective regime. If the conservative temperature and absolute salinity of the mixed layers above and below the interface both decrease with pressure, the interface belongs to the salt-finger regime. The algorithm detects more interfaces in the salt-finger regime (27 %) than in the diffusive-convective

regime (11 %, Fig. 6a). As expected, most interfaces with diffusive-convective characteristics have Turner angles between $-90° < Tu < -45°$ (blue histogram in Fig. 6a) and most salt-finger interfaces have Turner angles between $45° < Tu < 90°$ (red histogram in Fig. 6a). This implies that these interface properties are consistent with the background stratification.

### 3.5 Sequences of interfaces

The final step of the detection algorithm is to only select vertical sequences of at least two interfaces in the same double-

150 diffusive regime that are separated from each other by one mixed layer (Fig. 6b). This step removes most thermohaline intrusions, as these are characterized by alternating mixed layers in the diffusive-convective and salt-finger regime (Bebieva and Timmermans, 2017). In this final step, the algorithm also removes also salt-finger interfaces and diffusive-convective interfaces outside their favourable Turner angle (compare Fig. 6a and Fig. 6b).

After applying this final step of the algorithm, we obtain a global dataset consisting of 166,141 interfaces in the salt-finger

regime and 119,619 interfaces in the diffusive-convective regime. The distribution of the pressure levels and height of the

mixed layers at these interfaces is displayed in Fig. 1. In general, mixed layers in the diffusive-convective regime are found at lower pressure levels than mixed layers in the salt-finger regime (Fig. 1d). At the same time, the height of the mixed layers in the diffusive-convective regime are smaller, which is in line with previous observations (Fig. 1e, e.g., Radko, 2013). Recall that the algorithm required a minimal interface height of 2 dbar, which implies that, following equation 5, the minimal mixed layer height is 3 dbar and that the detection of interfaces is cut off below these limits. Consequently, the interfaces with smaller heights are missed by the algorithm. Figure 1e indicates that this is more problematic for interfaces in the diffusive-convective regime than for interfaces in the salt-finger regime.

Examples of thermohaline staircases, which were selected based on their high number of interfaces, are shown in Figure 7. In line with previous results (Rudels, 2015), staircases in the diffusive-convective regime (Fig. 7a) are mainly detected on the thermocline with the conservative temperature increasing with depth. These staircases are predominantly located in the Arctic Ocean at a depth between 300-400 m, which is between the warm and saline Atlantic Water and cold and fresh surface waters (Rudels, 2015). Figure 7a also indicates that the deepest mixed layer of some thermohaline staircases is located at the temperature maximum, which suggests that this lowest layer might be the result of thermohaline intrusions (Ruddick and Kerr, 2003). There, the algorithm identified a mixed layer, because temperature and salinity stratification were weak enough (see Section 3.1). Furthermore, both conservative temperature and absolute salinity in this mixed layer are larger than in the mixed layer above. While both are typical for a staircase in the diffusive-convective regime, the algorithm does not detect whether this mixed layer is a temperature maximum, which could indicate that arose from thermohaline intrusions. Note that this only concerns the deepest mixed layers of the staircases, and that only the characteristics of the interfaces in between mixed layers are labelled as part of a staircase by the algorithm.

Thermohaline staircases with a high number of steps in the salt-finger regime are detected on the main thermocline where the conservative temperature decreases with depth (Fig. 7b). Compared to the staircases in the diffusive-convective regime, these staircases are located slightly deeper at 400-700 m. While the locations of these staircases vary, they are located above the cold and fresh Antarctic Intermediate Water, which is observed below 700 m (Tsuchiya, 1989; Fine, 1993; Talley, 1996).

For each thermohaline staircase, characteristics of the interfaces and mixed layers, such as their conservative temperature, absolute salinity and height, are available in the dataset. An overview of the provided variables is given in Table A1. The detection algorithm is verified by comparing our data to independent observations in three regions in Section 5.

## 4    Robustness of the detection algorithm

The algorithm requires four input parameters: the moving average window, a threshold for the maximum density gradients of the mixed layers, the maximum density difference of the mixed layers and the maximum height of the interface (Table 2). In this section, the sensitivity of the algorithm to each input parameter is assessed (Fig. 8).

The moving average window is used by the algorithm to compute the thermal expansion coefficient ($\alpha$), the haline contraction coefficient ($\beta$) and the density ratio ($R_\rho$). We varied the moving average window between 50 dbar and 350 dbar to assess the

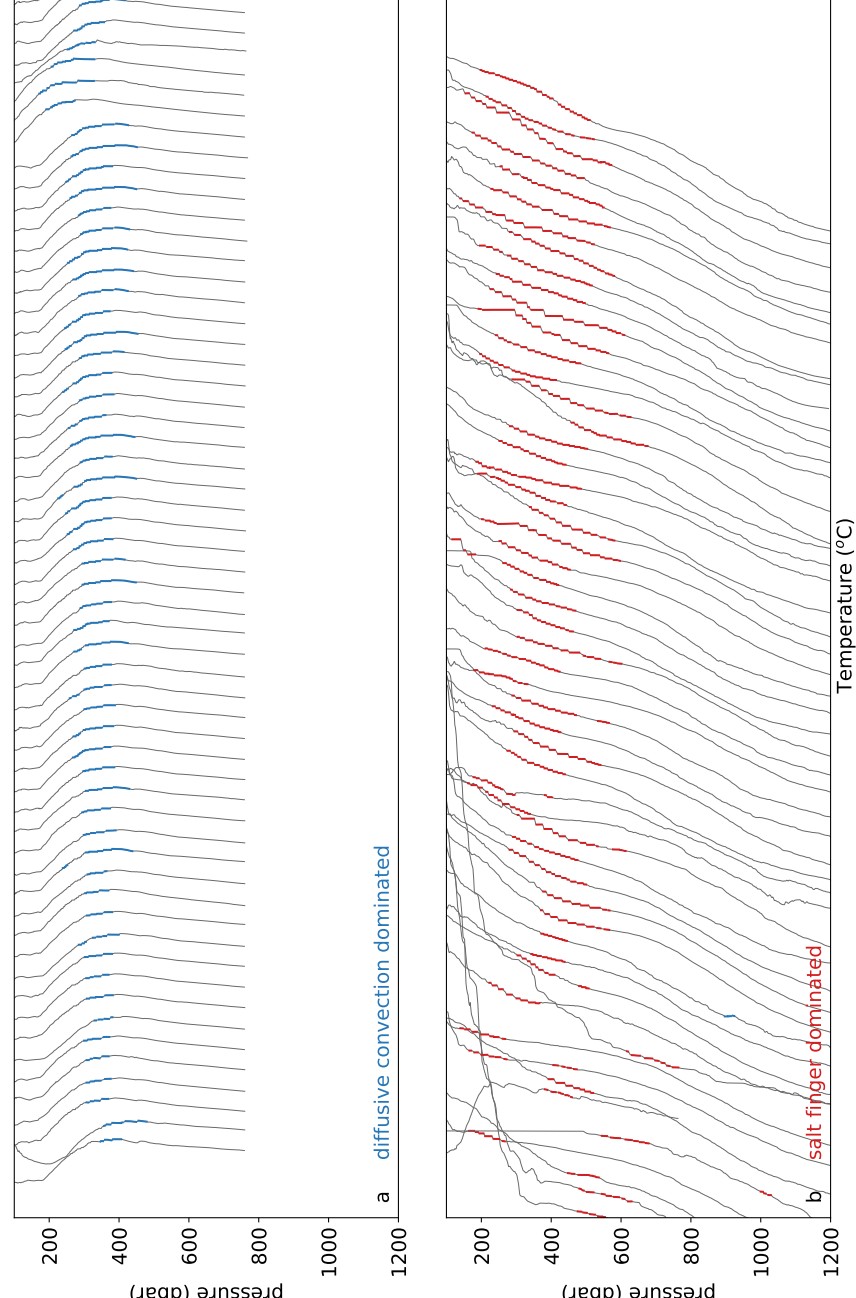

**Figure 7.** Example conservative temperature profiles selected by the staircase detection algorithm. They are ordered left-right by the number of steps detected. Top panel shows examples of increasing steps of diffusive convection, bottom panel shows examples of the salt-finger regime.

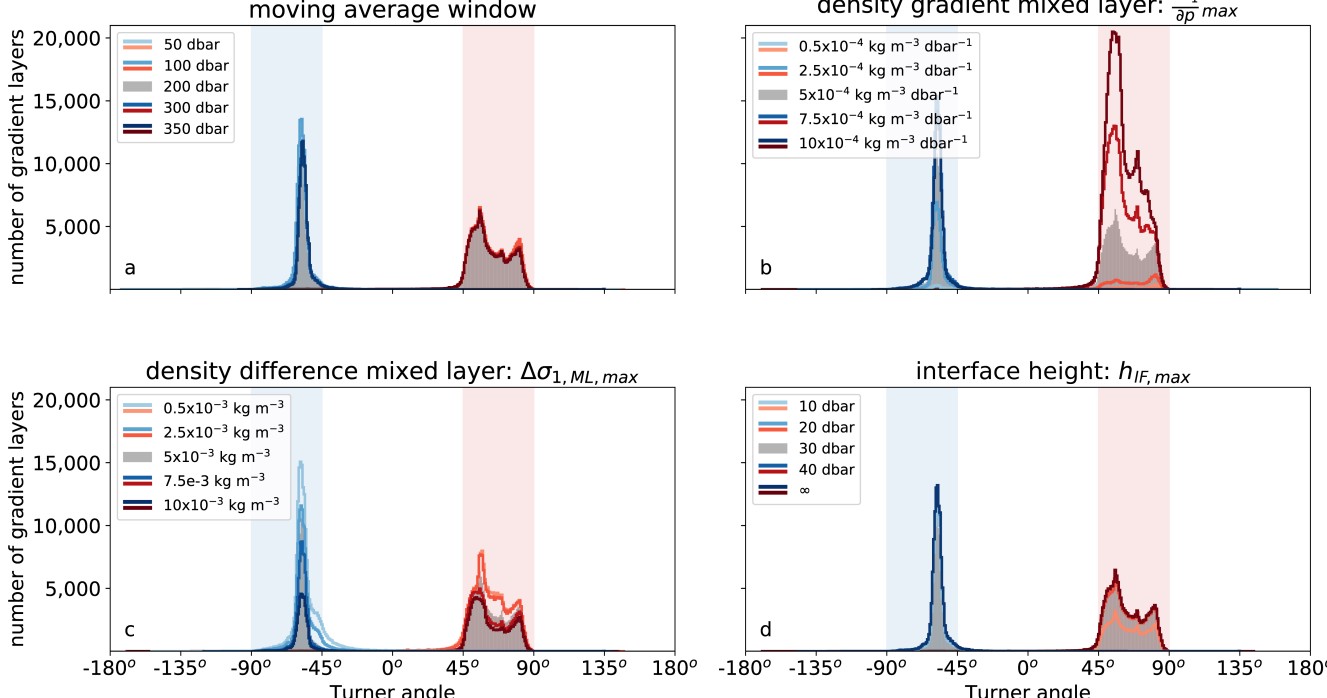

**Figure 8.** Number of detected interfaces obtained with the detection algorithm for different input parameters. Each subpanel shows the sensitivity of the detection algorithm to one input parameter: (a) moving average window, (b) density gradient of the mixed layer, (c) density difference within the mixed layer and (d) the maximum height of the interface. In each panel, the grey histogram corresponds to the default parameters listed in Table 2. The colored lines correspond to the varying parameter (see legend). Shaded regions indicate Turner angles in the diffusive-convective (blue) and salt-finger (red) regime.

sensitivity of the outcomes of this choice (Fig. 8a). We find that the varying moving average window does not result in large variations in detected mixed layers (Fig. 8a).

In contrast to the moving-average window, the detection algorithm is sensitive to the value set for the density gradient threshold of the mixed layer (Fig. 8b), which is used to obtain a reference pressure for the sub-surface mixed layers (Section 3.1). Not surprisingly, we detect more (less) interfaces when we increase (decrease) the allowed threshold density gradient. A small value allows for only the strongest mixed layers to be detected, which are usually referred to as well-defined staircases, while a large density gradient also allows for the detection of rough staircases (e.g., Durante et al., 2019). Although the number

of detected interfaces depends on the value set for this density gradient, the detected interfaces remain confined to the two double-diffusive regimes, indicating a robust outcome of the algorithm for the choice of this input parameter.

    Similar to the variations of the maximum density gradient, the variation of the maximum density difference allowed within the mixed layer results in a different number of detected interfaces (Fig. 8c). The number of detected mixed layers increases

when we decrease the maximum density difference allowed within the mixed layer. This effect is mostly visible in the diffusive-convective regime, as we obtained a decrease of 54 % of detected interfaces in the diffusive-convective regime compared an decrease of 31 % of detected interfaces in the salt-finger regime in case we doubled the density difference in the mixed layer ($\Delta\sigma_{1,max} = 10 \times 10^{-3}$ kg m$^{-3}$). This difference between the regimes is due to relatively small interface variations in the diffusive-convective regime compared to the salt-finger regime (Radko, 2013) and can be explained as follows: When a too large density difference is applied, the relatively small density gradients in the interfaces of the diffusive-convective regime are detected as mixed layers by the algorithm. Consequently, multiple mixed layers can be identified as a single mixed layer. However, if the applied density difference is too small, this could result in the detection of multiple mixed layers per staircase step.

The last input parameter of the detection algorithm concerns the interface height (Fig. 8d). As expected from Fig. 5b, variations of this input parameter do not result in large differences in the number of detected interfaces. If we omit this input parameter by setting it to infinity, we obtain a total increase of detected interfaces of 17 %.

Overall, the detection algorithm gives robust results as it predominantly detects interfaces within the double-diffusive regime (Fig. 8). In line with expectations, the detection algorithm is most sensitive to the threshold value for the maximum density gradient in the mixed layer and the density variations within the mixed layers. The four input variables allow for optimisation of the detection algorithm based on the regime and characteristics of the staircases.

## 5 Regional verification

The characteristics of thermohaline staircases obtained with the detection algorithm are compared to those obtained from previous observational studies for three major staircase regions: the Canada Basin in the Arctic Ocean, the Mediterranean Sea, and the C-SALT region in the tropical Atlantic Ocean. An overview is given in Tables 3-5.

In the Canada Basin (135°W-145°W, 75°N-80°N), the algorithm detects thermohaline staircases in the diffusive-convective regime in 90 % of the profiles (Table 3). Both the occurrence and depth range are comparable to what was reported by Timmermans et al. (2008) and Shibley et al. (2017), who analyzed thermohaline staircases from several Ice-Tethered Profilers, demonstrating that our detection algorithm indeed detects thermohaline staircases at the right location. Microstructure observations suggested that the thermohaline staircases in Canada Basin have interfaces heights of approximately $h_{IF}$ = 0.15 m (Padman and Dillon, 1987; Radko, 2013). Due to the vertical resolution of the profiles and the design of the algorithm (recall that the mixed layers are separated from each other by removing the upper and lower datapoint of the mixed layer, Section 3.1), the method is not capable of detecting very thin interfaces (Figure A1). As expected from these limitations for the detection of the interface heights, the algorithm detects conservative temperature and absolute salinity steps ($\Delta T_{IF}$ and $\Delta S_{IF}$, respectively) in the gradients layers that are in the upper ranges of earlier observations (Padman and Dillon, 1987; Timmermans et al., 2003, 2008; Shibley et al., 2017).

In the Mediterranean Sea, thermohaline staircases are characterized by relatively thick mixed layers that are separated by thick interfaces of up to 27 m (Zodiatis and Gasparini, 1996). In this region (0°E-15°E, 30°N-43°N), the detection algorithm

**Table 3.** Characteristics of thermohaline staircases in Canada Basin. The region of the global dataset is confined to: 135°W-145°W, 75°N-80°N. The observational techniques indicate if the data was obtained from Argo floats (Argo), Ice-Tethered Profilers (ITP), Conductivity Temperature Depth measurements (CTD) or microstructure measurements (MS). The dominant type of thermohaline staircases is indicated by DC (diffusive convection) and SF (salt-finger) with the percentage of occurrence between brackets. Ranges of the obtained variables of the global dataset are indicated by means of the 2.5 and 97.5-percentile.

| | technique | type | depth range (dbar) | $\Delta T_{IF}$ (°C) | $\Delta S_{IF}$ (g kg$^{-1}$) | $h_{IF}$ (dbar) |
|---|---|---|---|---|---|---|
| global dataset | ITP + Argo | DC (90 %) | 263 - 448 | 0.007 - 0.1 | 0.003 - 0.04 | 2 - 9 |
| Padman and Dillon (1987) | CTD+MS | DC (100 %) | 320 - 430 | 0.004 - 0.013 | 0.0016 - 0.0049 | 0.15 |
| Timmermans et al. (2003) | CTD | DC | 2400-2900 | 0.001 - 0.005 | 0.0035 - 0.0045 | 2 - 16 |
| Timmermans et al. (2008) | ITP | DC (96 %) | 200 - 300 | 0.04 | 0.014 | |
| Shibley et al. (2017) | ITP | DC (80 %) | | 0.04±0.01 | 0.01 ±0.003 | < 1 m |

**Table 4.** as Table 3, but for the Mediterranean Sea (0°E-15°E, 30°N-43°N).

| | technique | type | depth range (dbar) | $\Delta T_{IF}$ (°C) | $\Delta S_{IF}$ (g kg$^{-1}$) | $h_{IF}$ (dbar) |
|---|---|---|---|---|---|---|
| global dataset | ITP + Argo | SF (6 %) | 287 - 866 | 0.0097 - 0.12 | 0.0017 - 0.031 | 3 - 21 |
| Zodiatis and Gasparini (1996) | CTD | SF | 600 - 2500 | 0.04 - 0.17 | 0.01 - 0.04 | 2 - 27 |
| Bryden et al. (2014) | CTD | SF (32 %) | 600 - 1400 | 0.03 - 0.13 | 0.009 - 0.03 | 5 - 16 |
| Buffett et al. (2017) | seismic imaging | SF | 550 - 1200 | | | |
| Durante et al. (2019) | CTD | SF | 500 - 2500 | approx. 0.15 | | 4 -17 |

detected thermohaline staircases with interfaces up to 21 dbar in 6 % of the profiles, which is comparable to previous observations (Table 4). An example of the detection of a Mediterranean staircase is shown in Figure A2. We find that the depth at which the thermohaline staircases occur is underestimated by the detection algorithm. This could be explained by the fact that most Mediterranean observations are obtained by the Coriolis DAC (Fig. 1a). From this DAC, approximately 50 % of the profiles have observations that are deeper than 1000 dbar (Fig. 1b), which means that the coverage below 1000 dbar is limited in the Mediterranean Sea. Although the Argo floats, and consequently the detection algorithm, do not cover the full extent of the staircases (Fig. 1), the conservative temperature and absolute salinity steps that are found are similar to previous observations (Table 4). Note that the conservative temperature and absolute salinity steps of the staircases increase with depth (Zodiatis and Gasparini, 1996), which explains why the conservative temperature and absolute salinity steps detected by the algorithm are slightly smaller than those observed in the deeper observations (Zodiatis and Gasparini, 1996; Durante et al., 2019).

**Table 5.** as Table 3, but for the western tropical North Atlantic Ocean (53°W-58°W, 10°N-15°N).

| | technique | type | depth range (dbar) | $\Delta T_{IF}$ (°C) | $\Delta S_{IF}$ (g kg$^{-1}$) | $h_{IF}$ (dbar) |
|---|---|---|---|---|---|---|
| global dataset | ITP + Argo | SF (60 %) | 265 - 837 | 0.019 - 0.97 | 0.0014 - 0.16 | 3 - 18 |
| Schmitt et al. (1987) | CTD+MS | SF | 180 - 650 | 0.5 - 0.8 | 0.1 - 0.2 psu | 1 - 10 |
| Schmitt (2005) | CTD+MS | SF | 200 - 600 | <1 | | 0.5 - 5 |
| Fer et al. (2010) | Seismic imaging | SF | 550 - 700 | | | |

In the C-SALT region in the western tropical North Atlantic Ocean (53°W-58°W, 10°N-15°N), the algorithm detected thermohaline staircases in the salt-finger regime in 60 % of the profiles (Table 5). Similar to previous studies (Schmitt et al., 1987; Schmitt, 2005; Fer et al., 2010), the algorithm detected thermohaline staircases on the main thermocline (see example in Fig. A3). Again, the interface height is slightly overestimated by the detection algorithm, but the algorithm obtained conservative temperature and absolute salinity steps comparable to previous studies.

Overall, the comparison between the outcomes of the detection algorithm with previous studies indicates that the detection algorithm performs well. The small overestimation of the interface height can be attributed to the limited vertical resolution and the limitation imposed by the detection algorithm to avoid detection of false positives. Despite this overestimation, the interfaces are detected at the correct depths with conservative temperature and absolute salinity steps within realistic ranges. Therefore, we conclude that the detection algorithm is very suitable for the automated detection of thermohaline staircases in large and quickly growing datasets like the Argo float and Ice-Tethered Profilers data.

## 6 Conclusions

In this study, we presented an algorithm to automatically detect thermohaline staircases from Argo float profiles and Ice-Tethered Profiles. As these thermohaline staircases have different mixed layer heights and temperature and salinity steps across the interfaces in different staircase regions, the design of the detection algorithm is based on the typical vertical structure and shape of the staircases (Fig. 3-5). Note that by formulating the algorithm solely on this vertical structure of the staircases, we could use the Turner angle of the detected staircases for verification. Using this Turner angle, we showed that the structures are within the two double-diffusive regimes: the salt-finger regime and the diffusive-convective regime (Fig. 6).

We optimized the input of the algorithm such that it provides a global overview and limits the number of detected false positives. As a result, the regional verification in Section 5 indicated that the data pre-processing and data analysis have some limitations. For example, the vertical resolution of 1 dbar in the profiles is too course to capture all staircase steps in the Arctic Ocean. In the Mediterranean, the Argo floats did not dive deep enough to capture the full depth of the staircase region. However, the fact that (i) the algorithm detects thermohaline staircases at realistic depth ranges, with (ii) conservative temperature and

absolute salinity steps across the interfaces, and in (iii) the same double-diffusive regime as previous studies (Table 3-Table 5), indicates that the algorithm itself performs well. Therefore, when considering an individual staircase region, we recommend optimizing the input variables of the algorithm for that specific region and applying the algorithm on additional data, for example high-resolution CTD or microstructure profiles, where available.

A sensitivity analysis to different input parameters showed that the results of the detection algorithm are robust; the detected staircase interfaces are confined to the double-diffusive regimes. Furthermore, the comparison between the detected interface characteristics of thermohaline staircases in three prevailing staircase regions and previous observations, suggested that the detection algorithm accurately captures both double-diffusive regimes. The algorithm detected correct magnitudes of the conservative temperature and absolute salinity steps in the interfaces, which allows for adequate estimates of the effective diffusivity in thermohaline staircases.

The global dataset resulting from the detection algorithm contains properties and characteristics of both mixed layers and interfaces. Combined with their locations, this data allows for a statistical analysis of thermohaline staircases on global scales. For example, the global occurrence of thermohaline staircases could give insight in the contribution of double diffusion to the global mechanical energy budget. Moreover, the interface characteristics can be used to validate model and laboratory results on how double-diffusive mixing impacts the regional ocean circulation.

## 7   Code and data availability

Both algorithm and global dataset are available at doi: https://doi.org/10.5281/zenodo.4286170 (van der Boog et al., 2020). The algorithm is written in Python3 and is available under the Creative Commons Attribution 4.0 License. More details on the functions and output of the algorithm are depicted in Table A1 and Table A2, respectively. The structure of the algorithm is displayed in Figure A4.

*Author contributions.* CvdB and OK designed the detection scheme. CvdB wrote the paper and was supervised CK, JP and HD who helped shape the analysis and paper.

*Competing interests.* The authors declare that they have no conflict of interest.

*Acknowledgements.* The Ice-Tethered Profiler data were collected and made available by the Ice-Tethered Profiler Program (Krishfield et al., 2008; Toole et al., 2011) based at Woods Hole Oceanogaphic Institution (http://www.whoi.edu/itp). The Argo data were collected and made freely available by the International Argo Program and the national programs that contribute to it (http://www.argo.ucsd.edu, http://argo.jcommops.org). The Argo Program is part of the Global Ocean Observing System (Argo, 2020). The work of Carine van der Boog is financed by a Delft Technology Fellowship awarded to Caroline Katsman.

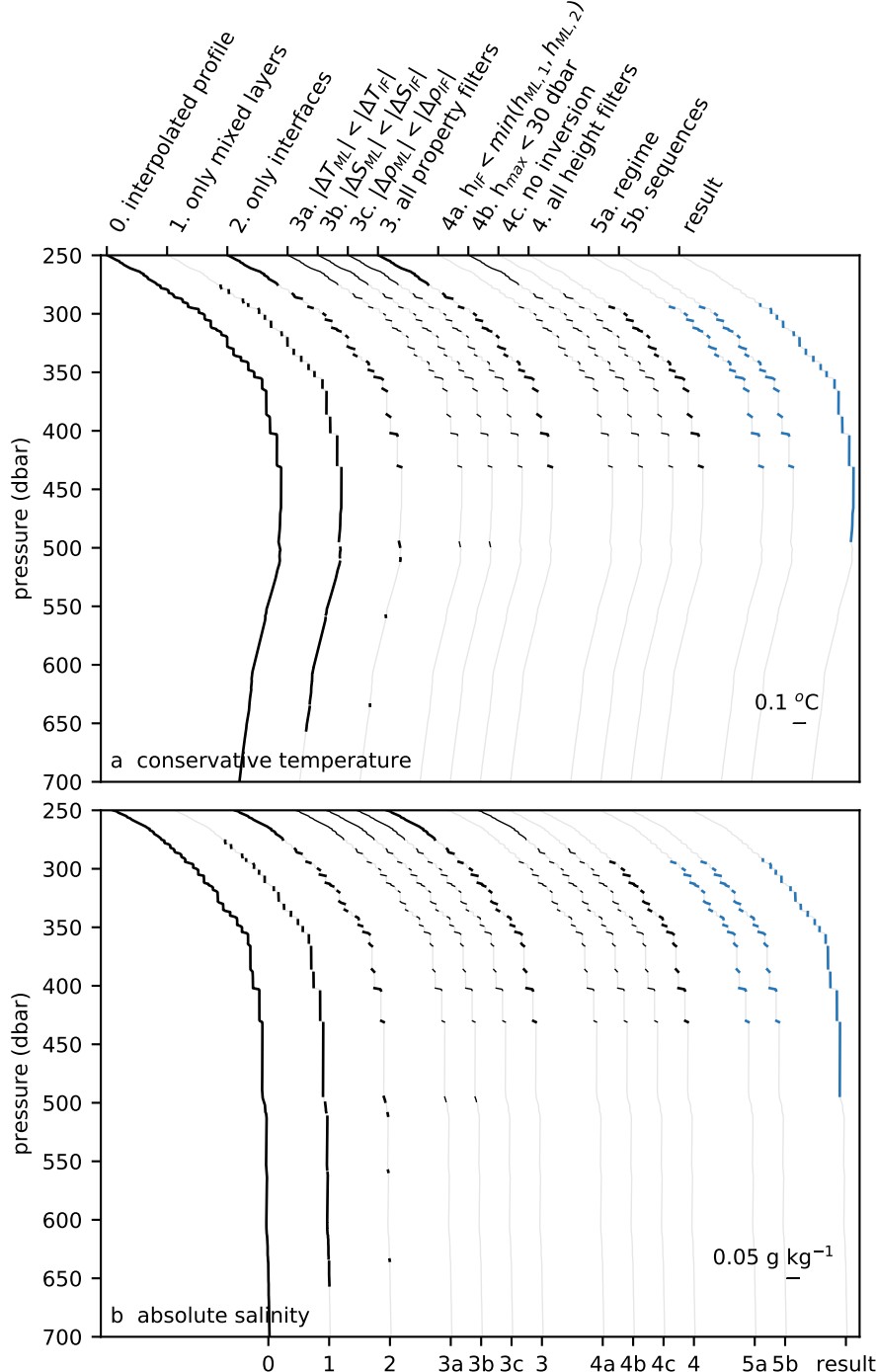

**Figure A1.** Steps of the detection algorithm applied on a profile in the Arctic Ocean, where steps are indicated on separate (a) conservative temperature and (b) absolute salinity profiles. Each profile is shifted for clarity. Similar to Figures 4-6, an interface is not considered by the detection algorithm when the interface characteristics did not meet the requirements of a previous step. Original profile is taken from Ice-Tethered-Profiler ITP64 at 137.8°W and 75.2°N on 29 January 2013. The details of the data preparation and the algorithm steps are discussed in Section 2 and Section 3, respectively.

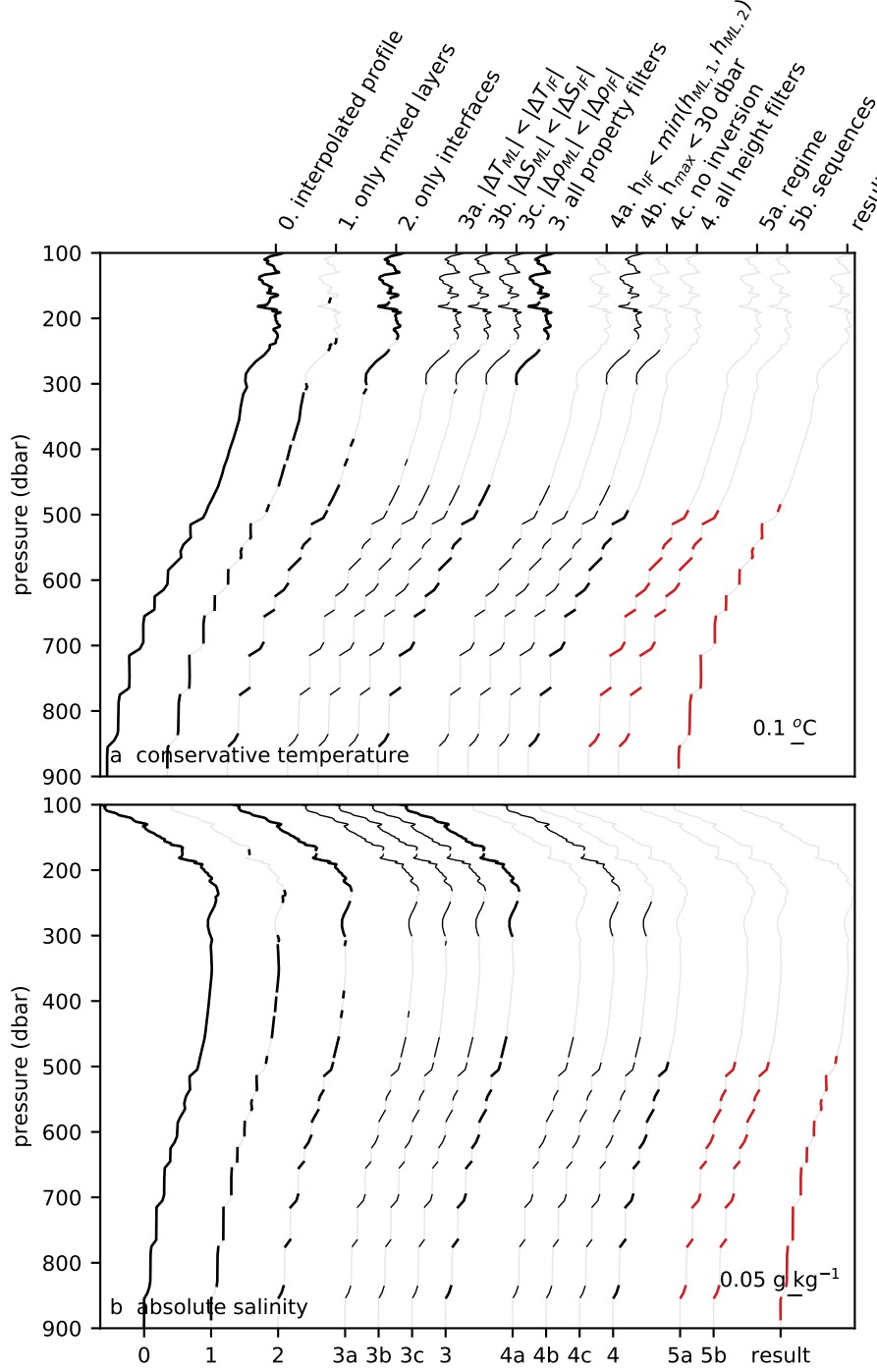

**Figure A2.** as Figure A1, but for a profile in the Mediterranean Sea. Original profile is taken from Argo float 6901769 at 8.9$^o$E and 37.9$^o$N on 31 October 2017.

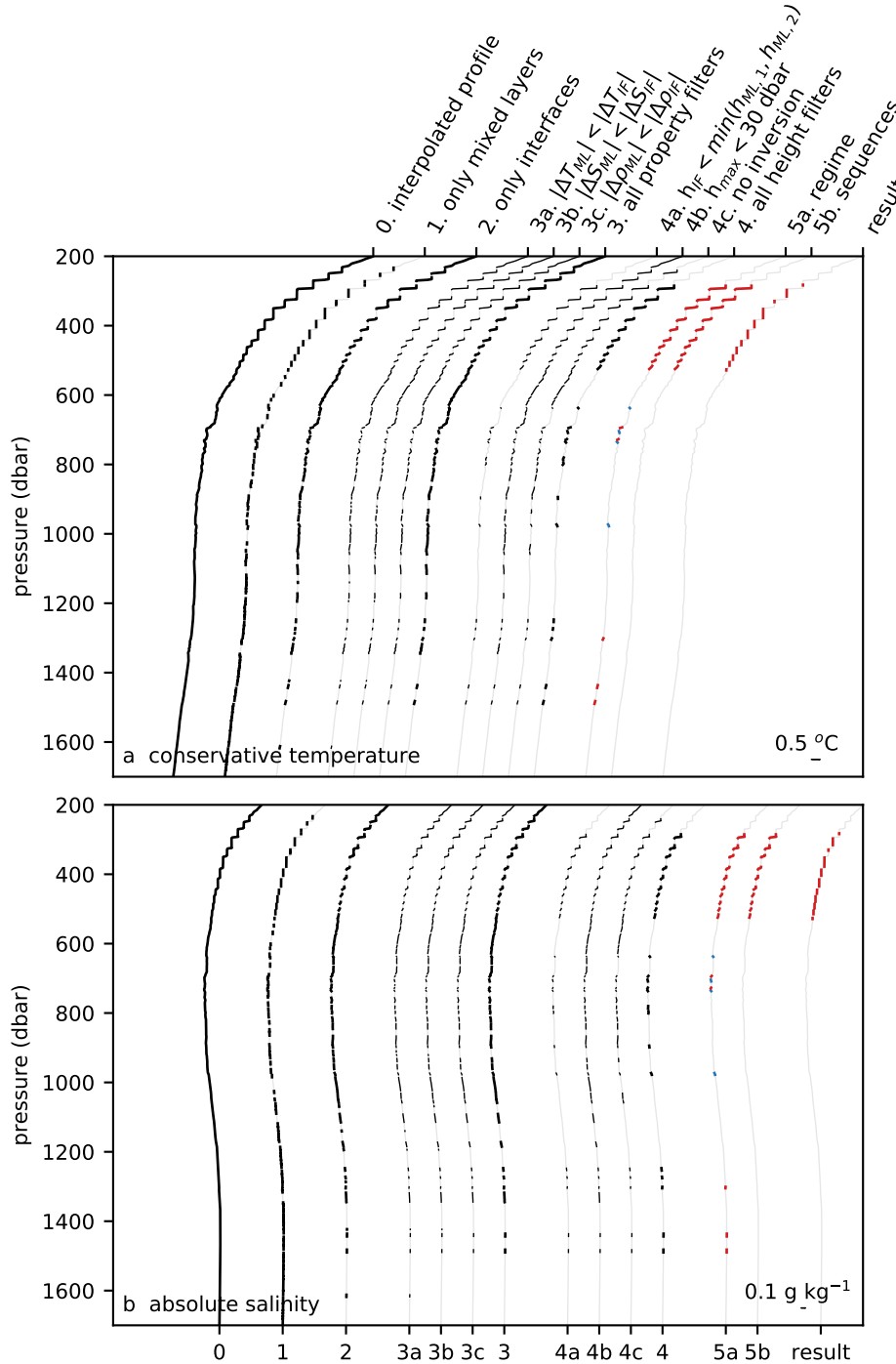

**Figure A3.** as Figure A1, but for a profile in the western tropical North Atlantic. Original profile is taken from Argo float 4901478 at 53.3°W and 11.6°N on 9 August 2014.

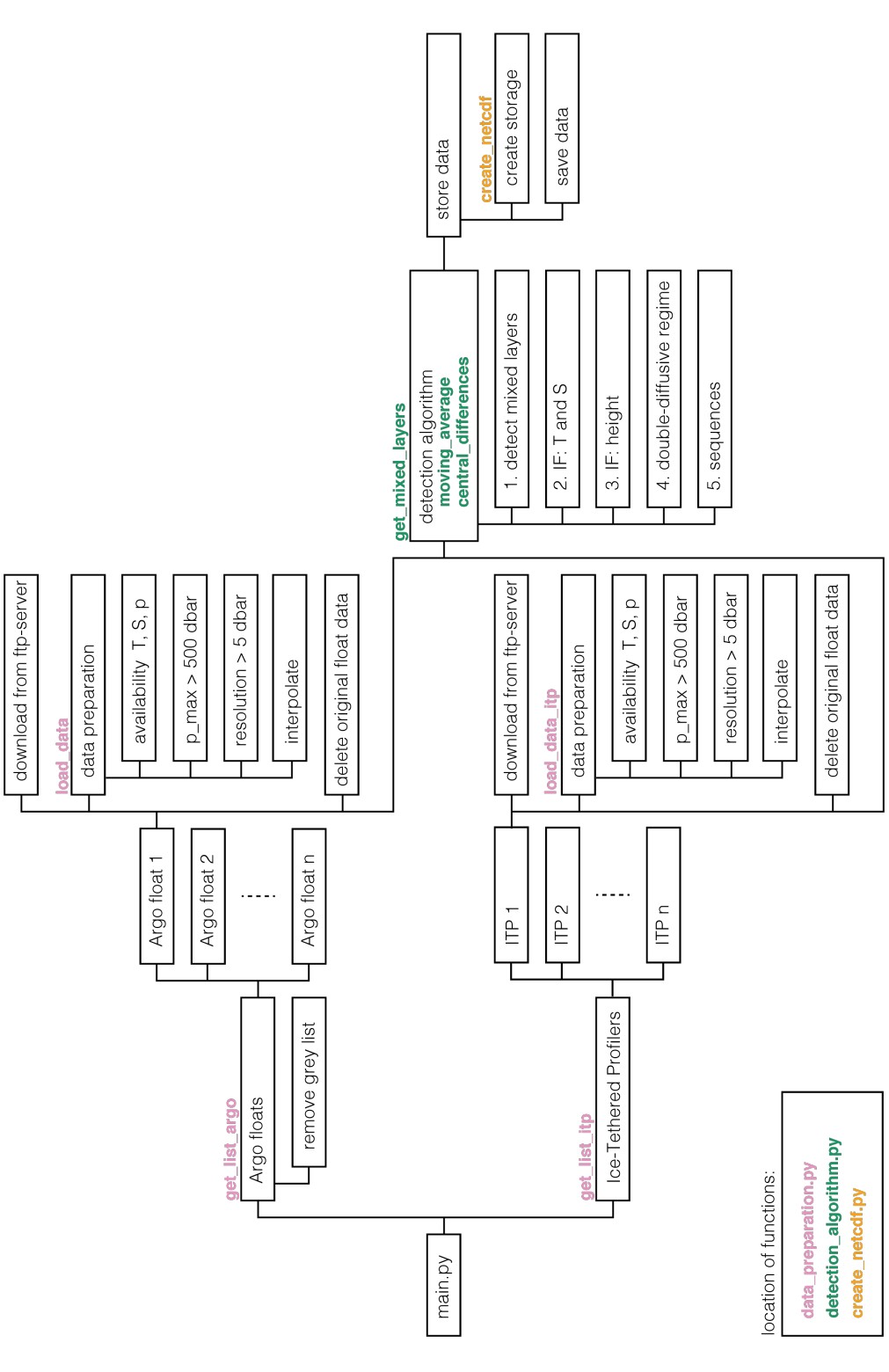

**Figure A4.** Structure of the software. Each step in the software is shown by a box. Whenever a particular step is contained inside a function, the name of the function is mentioned above the step. Details of the preprocessing of the data and the detection algorithm are discussed in Sections 2 and Section 3, respectively.

**Table A1.** Metadata of all variables that are saved in the dataset.

| variable | unit | description |
|---|---|---|
| floatID | | float identification number of ITP or Argo float |
| lat | °E | latitude of observation |
| lon | °N | longitude of observation |
| juld | days | Julian date of observation |
| ct | °C | conservative temperature (full profile) |
| sa | g kg$^{-1}$ | absolute salinity (full profile) |
| $\mathrm{ML}_{SF}$ | | mask with mixed layers in the salt-finger regime |
| $\mathrm{ML}_{DC}$ | | mask with mixed layers in the diffusive-convective regime |
| $\mathrm{p}_{ML}$ | dbar | average pressure of the mixed layer |
| $\mathrm{h}_{ML}$ | dbar | height of the mixed layer |
| $\mathrm{T}_{ML}$ | °C | average conservative temperature of mixed layer |
| $\mathrm{S}_{ML}$ | g kg$^{-1}$ | average absolute salinity of mixed layer |
| $\mathrm{Tu}_{ML}$ | ° | average Turner angle of mixed layer |
| $\mathrm{R}_{ML}$ | | average density ratio of the mixed layer |
| $\mathrm{h}_{IF}$ | dbar | height of the interface |
| $\mathrm{Tu}_{IF}$ | ° | Turner angle at the center of the interface |
| $\mathrm{R}_{IF}$ | | density ratio at the center of the interface |
| $\Delta\mathrm{T}_{IF}$ | °C | conservative temperature difference within the interface |
| $\Delta\mathrm{S}_{IF}$ | g kg$^{-1}$ | absolute salinity difference within the interface |

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

**Table A2.** Functions used in the software.

| function | input | output | description |
|---|---|---|---|
| get_list_argo | centers, filename | directory, floats, float_list | Access ftp server (ftp.ifremer.fr) and navigate through directories of the Data Assembly Centers (*centers*) to locate the Argo floats from the input list (*filename*). Directory of floats on the ftp-server are given in *directory*. The full list of Argo float before removal of the floats of the Grey list is given in *floats*. Argo floats mentioned on the Grey list are removed. Required packages: ftplib, numpy, pandas |
| get_list_itp | - | list of floats | Access ftp server (ftp.whoi.edu) to obtain list of available Ice-Tethered Profilers. Required packages: ftplib, numpy |
| load_data | filename, interp | p, lat, lon, ct, sa, juld | The profiles of a single Argo float (*filename*) are evaluated and linearly interpolated to a resolution of 1 dbar (*interp=True*). Only profiles with an average resolution is finer than 5 dbar and pressure levels exceeding 500 dbar are considered. Output contains interpolated data of pressure, latitude (*lat*), longitude (*lat*), conservative temperature (*ct*), absolute salinity (*sa*), Julian date (*juld*). Required packages: gsw, numpy, netCDF4, scipy |
| load_data_itp | path,profiles,interp | prof_no, p, lat, lon, ct, sa, juld | Similar as load_data, but then for Ice-Tethered Profilers. There is an additional output containing the FloatID of the ITP (*prof_no*). Required packages: datetime, gsw, numpy, pandas, scipy |
| get_mixed_layers | p, ct, sa, c1, c2, c3, c4 | ml, gl, masks | This is the detection algorithm. Input contains the pressure, conservative temperature, absolute salinity and the user-defined input parameters: $\partial\sigma_1/\partial p_{max}(c1)$, $\Delta\sigma_{1,ML,max}$ (*c2*), moving average window (*c4*), $h_{IF,max}$ (*c3*). The output are classes with the mixed layer characteristics (*ml*), interface characteristics (*gl*) and the masks (see Details in Table A1). Required packages: gsw, numpy, scipy. |
| moving_average2d | dataset, window | mav | Apply moving average window (*window*) to vertical profiles (*dataset*) obtain background profiles (*mav*). Required packages: numpy, scipy |
| central_differences2d | f, z | dfdz | Compute vertical gradients with central differences scheme. Required packages: numpy |