# Peer review of "Global dataset of thermohaline staircases obtained from Argo floats and Ice-Tethered Profilers"

_Earth System Science Data, 2020_

## Referee Comment (RC1) · Anonymous Referee #1 · 28 Aug 2020

General Comments: Excellent work, taking into account that the diapygnal fluxes at the interfaces of the stepped structures are considered comparable to those of the surface fluxes. The further use of the current results (in a next paper) can provide at global scale the contribution of the stepped structure diapygnal fluxes.

Specific Comments: A sentence is needed why only Argo floats and ice tethered profilers were used. Why not the high vertical resolution of the CTD profiles, which in most cases the vertical profiles are deeper than those of the floats. Before or after the Figure 6, add a sentence about the depth the stepped structures were detected, i.e. diffusive convection mostly was detected at depths between 300-400 m, while the salt finger

between 400-700 m.

Will be of interesting to see in a new paper using the CTD data from the SeaDataNet and or EMODNET portals applying the same technology of this ms to reveal the deepest stepped structures, as well as the fluxes estimates.

---

## Referee Comment (RC2) · Anonymous Referee #2 · 28 Sep 2020

This paper tackles the worthwhile problem of identifying and characterizing double-diffusive staircase structures in ocean temperature and salinity profiles. Unfortunately there are fundamental shortcomings of the work.

Without seeing representative profiles (from different regions), it is impossible to determine the extent to which the algorithm works. Figure 6b provides clues that it may be appropriate sometimes for the identification of salt-finger layers, although there are profile regions that appear to indicate steps which are not colored red (and it is unclear why). It would be helpful to be given some information about where the profiles are, and shown the detailed T-S structure. Figure 6a is clearly showing that the algorithm

is not working. The algorithm appears to have picked up the thermohaline intrusions underlying the double-diffusive staircase. One can see this immediately because of the regions that are deeper than the temperature maximum are marked blue. I would encourage the reviewers to examine some papers on the Arctic staircase and compare and validate their results against those. Similarly, the reader needs to see detailed profiles and validation. (As an aside, potential temperature should be used when examining step structures in deep water and the authors ought to compare profiles of potential temperature and temperature through deep staircases.)

The authors state that they only analyze profiles "with an average resolution finer than 5 dbar", and then that "all profiles are linearly interpolated to a vertical resolution of 1 dbar". First, it's unclear how interpolating to 1 dbar influences the results for a profile with a vertical resolution of $\sim$5dbar, and second (as the authors note) there are regions where gradient layer thicknesses are an order of magnitude smaller than the vertical resolution of their profiles. If this is not an issue, it will become clear when the authors show details of representative profiles as per my comment above.

---

## Referee Comment (RC3) · Anonymous Referee #3 · 30 Oct 2020

This paper describes the creation of a novel dataset to study thermohaline staircases in the ocean. It is a great example of how something new can be brought out of a widely-used dataset through a suitable data processing technique. The data processing is careful and well documented, and compares favorably against earlier regional studies. In particular, Figure 5 is impressive, where the authors appear to capture the salt-fingering and double-diffusive convection regimes based on the application of their straightforward criteria. The dataset created by the authors is quite unique and will undoubtedly be of use to others, particularly since it is distributed together with the software. I believe it should be published with minor revisions.

[Figure]

There are a few points I would like the authors to address.

– What is the estimated precision of the salinity, temperature, and density measurements, and how does this compare with typical step sizes? I ask because, if the precisions are coarse, or upstream rounding or truncation has been applied, a jump-like effect mimicking staircases could arise as an artifact. Here I think it is important to explicitly examine the measurement precisions and noise levels to rule out this possibility, rather than to simply argue that the final product seems to be physically meaningful.

– As the software is an important part of this contribution, I think it should be described in more detail, with language, license, and function or function names listed, together with a description of how the software is to be used and possibly listing inputs and outputs. It is important that the software is arranged as a function or functions rather than as a script, if it is to be useful to others.

–I find it conspicuous that, zooming on on Fig. 6a, I see a lot of staircases that appear to have been missed, lying just above the blue curves showing detections. Please discuss these and whether or not they are 'false negatives' that the method should detect but does not, and if they are then explain why such false negatives are acceptable.

–The problem that the authors examine is a difficult one. I am not sure that the most elegant solution has been found, as it is dependent upon the choices of a number of free parameters. Ideally, one should not have to specify a prior cutoffs; it would be preferable for these to emerge from the data based on examining statistical distributions. However, a parameter-free version of this product would probably take a great deal of more work and possibly different methods (e.g., least squares fits, statistical tests, etc.), and it is much better to have a satisfactory solution than none at all.

Because the authors have thought a lot about this problem, they are in a good position to describe the shortfalls of the current method and how it might be improved in the future. This would be a great topic to discuss at the end of the paper.

[Figure]

Minor comments

p 1, first paragraph, and p 2 line 31, "double-diffusive" should be hyphenated

p 1, line 14, "two orders of magnitude"

p 1, line 17, and p 5, line 93, "of the order"

p 1, line 19, "the the"

p 2, line 35, would recommend present tense

p 2, line 47, what is the gray list and where can it be found?

p 3, line 57, this is a second moving average, yes?

p 3, eqn 1, what is the meaning of the overbar?

p 4, lines 64, "the properties of any layer lying between" would be better

p 4, lines 74,75, and 76, "criterium" should be "criterion"

p 7, I believe the first paragraph is unnecessarily repeated

p 10, where are these example profiles from?

p 11 "optimalization" should be "optimization"

p 13, line 219–220, I am not sure what is being meant here. It seems a lot of physical assumptions have been made that are implicit in the parameter choices.

p 14, line 225 should say "both double-diffusive regimes" I believe

Table A1, Julian should be capitalized and density should not be

Many of the references have incorrectly capitalized titles or journal names.

---

## Author Comment (AC1) · 25 Nov 2020

We would like to thank the reviewer for the time and effort spent reading our manuscript and for the useful comments and suggestions. A detailed response to all comments can be found below, where the black text indicates comments of the reviewer. The blue text denotes our response to these comments; line numbers refer to the revised version of the paper.

**Comments by the reviewer:**

General Comments: Excellent work, taking into account that the diapygnal fluxes at the interfaces of the stepped structures are considered comparable to those of the surface fluxes. The further use of the current results (in a next paper) can provide at global scale the contribution of the stepped structure diapygnal fluxes.

We indeed are working on a next paper where we compute the contribution of doublediffusive processes to the global mechanical energy budget. Furthermore, we noticed that 'interfaces' is a more widely accepted term than gradient layers. Therefore, we replaced all mentions of 'gradient layers' by 'interfaces' throughout the manuscript. This included replacements in Figure 4, 5, 6, and 8.

Specific Comments: A sentence is needed why only Argo floats and ice tethered profilers were used. Why not the high vertical resolution of the CTD profiles, which in most cases the vertical profiles are deeper than those of the floats.

We limited ourselves to Argo floats and Ice-Tethered Profilers, because they have a global coverage and we could use them to show that the algorithm performs its task well. However, we agree with the reviewer that it would be interesting to extend the dataset with more data in the future. We also added a sentence to the conclusions to highlight this possibility.

**Lines 266-268:**

'Therefore, when considering an individual staircase region, we recommend optimizing the input variables of the algorithm for that specific region and applying the algorithm on additional data, for example high-resolution CTD or microstructure profiles, where available.'

Furthermore, we added additional information about the original vertical resolution of the profiles used in this study (Fig. 1 and Table 1). We added the Figure and Table at the end of this response.

**Lines 54-58:**

'Details on the origin and vertical resolution of the profiles are depicted in Table 1 and Figure 1, in which Figure 1b confirms that all profiles have observations deeper than 500 dbar. Furthermore, the average vertical resolution of the profiles indicates the average resolution is well below the 5 dbar that was used as a threshold (Fig. 1c). After this quality control, 487,493 vertical temperature and salinity profiles remain.'

Before or after the Figure 6, add a sentence about the depth the stepped structures were detected, i.e. diffusive convection mostly was detected at depths between 300-400 m, while the salt finger between 400-700 m.

We rewrote the paragraph to clarify that we selected the staircases with most steps. In addition, we indicated the water masses between which these staircases are found (and provided references):

**Lines 164-167:**

'In line with previous results (Rudels, 2015), staircases in the diffusive-convective regime (Fig. 7a) are mainly detected on the thermocline with the conservative temperature increasing with depth. These staircases are predominantly located in the Arctic Ocean at a depth between 300-400 m, which is between the warm and saline Atlantic Water and cold and fresh surface waters (Rudels, 2015)'

**Lines 175-178:**

'Thermohaline staircases with a high number of steps in the salt-finger regime are detected on the main thermocline where the conservative temperature decreases with depth (Fig. 7b). Compared to the staircases in the diffusive-convective regime, these staircases are located slightly deeper at 400-700 m. While the locations of these staircases vary, they are located above the cold and fresh Antarctic Intermediate Water, which is observed below 700 m (Tsuchiya, 1989; Fine, 1993; Talley, 1996).

Will be of interesting to see in a new paper using the CTD data from the SeaDataNetand or EMODNET portals applying the same technology of this ms to reveal the deepest stepped structures, as well as the fluxes estimates.

We agree with the reviewer that it would be interesting to use the algorithm on different datasets as well, but this is outside the scope of the present paper. No changes in text.

*Table 1* Number of floats and profiles in the global dataset. Profiles taken with Argo floats are categorised by the Data Assembly Center (DAC). Profiles taken with Ice-Tethered Profilers are categorised as ITP. The percentage between brackets indicates the relative contribution to the total number of profiles in the global dataset (487,493 profiles). More details on abbreviations of DAC can be found in Argo (2019)

| DAC      | Number of floats | profiles        |
|----------|------------------|-----------------|
| aoml     | 2692             | 312,285 (64.1%) |
| bodc     | 93               | 11,092 (2.3%)   |
| coriolis | 347              | 27,134 (5.6%)   |
| csio     | 81               | 15,099 (3.1%)   |
| csiro    | 378              | 42,942 (8.8%)   |
| incois   | 65               | 4,363 (0.9%)    |
| jma      | 205              | 22,919 (4.7%)   |
| kma      | 1                | 1 (0.0%)        |
| kordi    | 0                | 0 (0.0%)        |
| meds     | 145              | 9,285 (1.9%)    |
| nmdis    | 0                | 0 (0.0%)        |
| ITP      | 82               | 42,373 (8.7%)   |

---

## Author Comment (AC2) · 25 Nov 2020

*We would like to thank the reviewer for the time and effort spent reading our manuscript, and for the comments which have improved the manuscript significantly. A detailed response to all comments can be found below, where the blue text indicates our response to the reviewers' comments, which are denoted in black. Line numbers correspond to the revised manuscript.*

**Comments by the reviewer:**
This paper tackles the worthwhile problem of identifying and characterizing double-diffusive staircase structures in ocean temperature and salinity profiles. Unfortunately there are fundamental shortcomings of the work.

Without seeing representative profiles (from different regions), it is impossible to determine the extent to which the algorithm works. Figure 6b provides clues that it maybe appropriate sometimes for the identification of salt-finger layers, although there are profile regions that appear to indicate steps which are not colored red (and it is unclear why). It would be helpful to be given some information about where the profiles are, and shown the detailed T-S structure.

*The algorithm does detect thermohaline staircases not only using profiles of conservative temperature, but also using potential density and absolute salinity. Therefore, it is not always clear from conservative temperature profiles why a step is disregarded. To be more transparent about this selection, we added 3 figures in the Appendix of the revised paper with representative profiles of three well-known formation regions: the Arctic Ocean, the Mediterranean Sea, and the western tropical Atlantic Ocean. In these figures, we show the different steps of the algorithm. We also added the figures at the end of this reply.*

Figure 6a is clearly showing that the algorithm is not working. The algorithm appears to have picked up the thermohaline intrusions underlying the double-diffusive staircase. One can see this immediately because of the regions that are deeper than the temperature maximum are marked blue. I would encourage the reviewers to examine some papers on the Arctic staircase and compare and validate their results against those. Similarly, the reader needs to see detailed profiles and validation.

*Apparently, the algorithm was not clearly explained in the original paper and we use this comment to better explain the working and results of the algorithm (below and in the revised paper).*

*The algorithm detects stepped structures from vertical profiles of conservative temperature and absolute salinity. This implies that the algorithm can also detect mixed layers arising from thermohaline intrusions. Therefore, we added a paragraph to the introduction to discuss the origin of thermohaline staircases:*

> *Lines 17-23:*
> *'It is still a topic of discussion how double-diffusive convection leads to the formation of thermohaline staircases in oceanic environments (Merryfield, 2000). For example, Stern (1969) argued that small-scale mixing processes trigger the formation of internal waves. On the other hand, variations in the turbulent heat and salt fluxes (Radko, 2003) or in the counter-gradient buoyancy fluxes that sharpen density gradients (Schmitt, 1994) could also lead to the formation of thermohaline staircases.*

*Lastly, subsurface mixed layers can also arise from thermohaline intrusions (Merryfield, 2000). Although it remains unclear how these staircases arise, these studies agree that the formation of these subsurface mixed layers are related to double-diffusive processes.'*

*We also added a sentence to highlight the benefit of using a detection algorithm based on the vertical structure, such that the Turner angle can be used for validation:*

*Lines 74-75:*
*'The benefit of using the vertical structure, instead of using assumptions based on the Turner angle, is that we can use this angle to verify the results.'*

*We want to emphasize that our results show that most detected staircases are within double-diffusive regimes (Fig. 6). This suggests that we predominantly detect double-diffusive thermohaline staircases. However, similar to any other detection of thermohaline staircases, we cannot determine whether the origin of a subsurface mixed layer in double-diffusive regimes arises from thermohaline intrusions or from double-diffusive mixing. We added a paragraph to Section 3 and rephrased two sentences in the abstract and introduction to clarify this.*

*Line 1:*
*'Thermohaline staircases are associated with double-diffusive mixing.'*

*Lines 12-14:*
*'They are associated with double-diffusive processes, which in turn result from a two orders of magnitude difference between the molecular diffusivity of heat and that of salt (Stern, 1960).'*

*Line 167-174:*
*'Figure 7a also indicates that the deepest mixed layer of some thermohaline staircases is located at the temperature maximum, which suggests that this lowest layer might be the result of thermohaline intrusions (Ruddick and Kerr, 2003). There, the algorithm identified a mixed layer, because temperature and salinity stratification were weak enough (see Section 3.1). Furthermore, both conservative temperature and absolute salinity in this mixed layer are larger than in the mixed layer above. While both are typical for a staircase in the diffusive-convective regime, the algorithm does not detect whether this mixed layer is a temperature maximum, which could indicate that arose from thermohaline intrusions. Note that this only concerns the deepest mixed layers of the staircases, and that only the characteristics of the interfaces in between mixed layers are labelled as part of a staircase by the algorithm.'*

*Furthermore, we would like to note that it is difficult to design a staircase detection algorithm that is optimized for all staircase regions, due to large variations in the height of the mixed layers and temperature and salinity steps of the interfaces. In this global dataset, we aimed to optimize the global detection, such that we detect thermohaline staircases in all well-known formation regions. To show this in a transparent way, we added figures of representative profiles (Figure A1, A2, A3), and added a paragraph to the conclusions to discuss this issue.*

*Lines 260-268:*

*'We optimized the input of the algorithm such that it provides a global overview and limits the number of detected false positives. As a result, the regional verification in Section 5 indicated that the data pre-processing and data analysis have some limitations. For example, the vertical resolution of 1 dbar in the profiles is too course to capture all staircase steps in the Arctic Ocean. In the Mediterranean, the Argo floats did not dive deep enough to capture the full depth of the staircase region. However, the fact that (i) the algorithm detects thermohaline staircases at realistic depth ranges, with (ii) conservative temperature and absolute salinity steps across the interfaces, and in (iii) the same double-diffusive regime as previous studies (Table 3-Table 5), indicates that the algorithm itself performs well. Therefore, when considering an individual staircase region, we recommend optimizing the input variables of the algorithm for that specific region and applying the algorithm on additional data, for example high-resolution CTD or microstructure profiles, where available.'*

(As an aside, potential temperature should be used when ex-amining step structures in deep water and the authors ought to compare profiles of potential temperature and temperature through deep staircases.)

*It is not entirely clear to us why the reviewer insists that potential temperature should be used when examining step structures in deep water. We prefer to use conservative temperature over potential temperature, because thermohaline staircases are predominantly studied for their heat and salt fluxes through the interfaces. In contrast to potential temperature, conservative temperature can be regarded as a conservative variable and can be accurately used for computations regarding the heat content (Graham and McDougall, 2013). For further details on the conservative temperature, we refer to Graham and McDougall (2013):*

> *Graham, F. S., & McDougall, T. J. (2013). Quantifying the nonconservative production of Conservative Temperature, potential temperature, and entropy. Journal of Physical Oceanography, 43(5), 838-862. https://doi.org/10.1175/JPO-D-11-0188.1*

*To clarify that we use conservative temperature instead of potential temperature, we replaced 'temperature' by 'conservative temperature' and 'salinity' by 'absolute salinity' throughout the manuscript.*

*Furthermore, we added a sentence to motivate the usage of conservative temperature:*

> *Line 63-64:*
> *'Note that we use conservative temperature as this is more accurate than potential temperature in computations concerning heat fluxes and heat content (Graham and McDougall, 2013).'*

[Figure]

*Figure A 1 Steps of the detection algorithm applied on a profile in the Arctic Ocean, where steps are indicated on separate (a) conservative temperature and (b) absolute salinity profiles. Each profile is shifted for clarity. Similar to Figures 3-5, an interface is not considered by the detection algorithm when the interface characteristics did not meet the requirements of a previous step. Original profile is taken from Ice-Tethered-Profiler ITP64 at 137.8°W and 75.2°N on 29 January 2013. The details of the pre-processing and the algorithm steps are discussed in Section 2 and Section 3, respectively.*

[Figure]

*Figure A 2 as Figure A1, but for a profile in the Mediterranean Sea. Original profile is taken from Argo float 6901769 at 8.9ºE and 37.9ºN on 31 October 2017.*

[Figure]

*Figure A 3 as Figure A1, but for a profile in the western tropical North Atlantic. Original profile is taken from Argo float 4901478 at 53.3ºW and 11.6ºN on 9 August 2014.*

---

## Author Comment (AC3) · 25 Nov 2020

*We would like to thank the reviewer for the time and effort spent reading our manuscript, and for the comments which have improved the manuscript significantly. A detailed response to all comments can be found below, where the blue text indicates our response to the reviewers' comments, which are denoted in black. Line numbers correspond to the revised manuscript.*

**Comments by the reviewer:**

This paper describes the creation of a novel dataset to study thermohaline staircases in the ocean. It is a great example of how something new can be brought out of a widely-used dataset through a suitable data processing technique. The data processing is careful and well documented, and compares favorably against earlier regional studies. In particular, Figure 5 is impressive, where the authors appear to capture the salt-fingering and double-diffusive convection regimes based on the application of their straightforward criteria. The dataset created by the authors is quite unique and will undoubtedly be of use to others, particularly since it is distributed together with the software. I believe it should be published with minor revisions.

There are a few points I would like the authors to address.

– What is the estimated precision of the salinity, temperature, and density measurements, and how does this compare with typical step sizes? I ask because, if the precisions are coarse, or upstream rounding or truncation has been applied, a jump-like effect mimicking staircases could arise as an artifact. Here I think it is important to explicitly examine the measurement precisions and noise levels to rule out this possibility, rather than to simply argue that the final product seems to be physically meaningful.

*The accuracy of a temperature measurement in an Argo float or Ice-Tethered Profiler is $0.001^oC$; for salinity this is 0.001 psu. These errors are much smaller than typical temperature and salinity differences characterizing staircases and hence roundoff due to measurement error does not play a role in step detection.*

– As the software is an important part of this contribution, I think it should be described in more detail, with language, license, and function or function names listed, together with a description of how the software is to be used and possibly listing inputs and outputs. It is important that the software is arranged as a function or functions rather than as a script, if it is to be useful to others.

*We thank the reviewer for this suggestion. We added a figure with the structure of the software and a table with the separate functions of the software (Table A2). The figure with the structure of the software is also added at the end of this reply. We have added the license and language at the code availability.*

> *Lines 281-284:*
> *'Both algorithm and global dataset are available at doi: https://doi.org/10.5281/zenodo.4286170 (van der Boog et al., 2020). The algorithm is written in Python3 and is available under the Creative Commons Attribution 4.0 License. More details on the functions and output of the algorithm are depicted in Table A1 and Table A2, respectively. The structure of the algorithm is displayed in Figure A4. '*

–I find it conspicuous that, zooming on on Fig. 6a, I see a lot of staircases that appear to have been missed, lying just above the blue curves showing detections. Please discuss these and whether or not they are 'false negatives' that the method should detect but does not, and if they are then explain why such false negatives are acceptable.

*We agree with the reviewer that it is not entirely clear from Figure 7a why some mixed layers are missed by the algorithm. A small part of these mixed layers is missed due to the resolution of the original profiles. We have clarified this in the text.*

> *Lines 224-226:*
> *'Due to the vertical resolution of the profiles and the design of the algorithm (recall that the mixed layers are separated from each other by removing the upper and lower datapoint of the mixed layer, Section 3.1), the method is not capable of detecting very thin interfaces (Figure A1).'*

*The other part of the mixed layers is missed because the algorithm detects thermohaline staircases not only using profiles of conservative temperature (as shown in Figure 7a), but also using potential density and absolute salinity. Therefore, it is not always clear from conservative temperature profiles why a step is disregarded. To be more transparent about this selection, we added 3 figures in the Appendix of the revised paper with representative profiles of three well-known formation regions: the Arctic Ocean, the Mediterranean Sea, and the western tropical Atlantic Ocean. In these figures, we show the different steps of the algorithm. We also added the figures at the end of this reply.*

–The problem that the authors examine is a difficult one. I am not sure that the most elegant solution has been found, as it is dependent upon the choices of a number of free parameters. Ideally, one should not have to specify a prior cutoffs; it would be preferable for these to emerge from the data based on examining statistical distributions. However, a parameter-free version of this product would probably take a great deal of more work and possibly different methods (e.g., least squares fits, statistical tests, etc.), and it is much better to have a satisfactory solution than none at all.

*Yes, we agree with the reviewer. The algorithm mainly depends on the parameters to detect the mixed layers (Fig. 8). It would indeed be more elegant to remove all parameters from the algorithm, but this is outside the scope of this paper.*

Because the authors have thought a lot about this problem, they are in a good position to describe the shortfalls of the current method and how it might be improved in the future. This would be a great topic to discuss at the end of the paper.

*The major shortfall of the algorithm is the preprocessing of the data and, consequently, the vertical resolution. We now discuss this shortfall, and how to resolve it, in the revised text.*

> *Lines 260-268:*
> *'We optimized the input of the algorithm such that it provides a global overview and limits the number of detected false positives. As a result, the regional verification in Section 5 indicated that the data pre-processing and data analysis have some limitations. For example, the vertical resolution of 1 dbar in the profiles is too course to capture all staircase steps in the Arctic Ocean. In the Mediterranean, the Argo*

*floats did not dive deep enough to capture the full depth of the staircase region. However, the fact that (i) the algorithm detects thermohaline staircases at realistic depth ranges, with (ii) conservative temperature and absolute salinity steps across the interfaces, 265 and in (iii) the same double-diffusive regime as previous studies (Table 3-Table 5), indicates that the algorithm itself performs well. Therefore, when considering an individual staircase region, we recommend optimizing the input variables of the algorithm for that specific region and applying the algorithm on additional data, for example high-resolution CTD or microstructure profiles, where available.'*

Minor comments
p 1, first paragraph, and p 2 line 31, "double-diffusive" should be hyphenated

*Corrected throughout the manuscript. Following the same grammar rule, we replaced Ice Tethered Profilers by Ice-Tethered Profilers.*

p 1, line 14, "two orders of magnitude"

*Corrected (line 13).*

p 1, line 17, and p 5, line 93, "of the order"

*Corrected (line 16, line 106).*

p 1, line 19, "the the"

*Corrected (line 24).*

p 2, line 35, would recommend present tense

*We agree, we changed the tense.*

p 2, line 47, what is the gray list and where can it be found?

*The gray list is a list of Argo floats that have problems with one or more sensors. We have mentioned this in the revised manuscript:*

> *Lines 51-53:*
> *'First a quality check is performed, where a profile is excluded from analysis if it was taken by an Argo float mentioned on the grey list. This grey list contains floats that may have problems with at least one of the sensors (https://www.nodc.noaa.gov/argo/grey_floats.htm).'*

p 3, line 57, this is a second moving average, yes?

*No, this is a first moving average, instead of the 200 dbar. We have clarified this in the text.*

> *Lines 67-68:*
> *'The Turner angle is computed using profiles that were smoothed with a moving average of 50 dbar instead of 200 dbar'*

p 3, eqn 1, what is the meaning of the overbar?

*The overbar indicated that the temperature and salinity profiles were smoothed. We understand that this is unclear, and the overbar is not necessary. Therefore, we decided to remove the overbar from equation 1.*

p 4, lines 64, "the properties of any layer lying between" would be better

*We thank the reviewer for this suggestion. We rephrased the sentence:*

> *Lines 78-79:*
> *'Next, the properties of any layer lying between the mixed layers (the interfaces, IF, orange dots in Fig. 3) are assessed by applying a minimum in temperature and salinity variations.'*

p 4, lines 74,75, and 76, "criterium" should be "criterion"

*Corrected (lines 86, 87, and 90).*

p 7, I believe the first paragraph is unnecessarily repeated

*Yes, we agree. We have changed the first paragraph and removed all repetitions.*

> *Lines 129-133:*
> *'Furthermore, the tallest observed interfaces are found in the Mediterranean Sea with heights up to $h_{IF} = 27$ m, where they separate mixed layers of over 100 m (Zodiatis and Gasparini, 1996; Radko, 2013). To prevent false detection of large vertical interfaces of up to hundreds of meters, we limit the interface height to $h_{IF,max} = 27$ dbar (Table 2, Fig. 5b). This only affects the classification of 1 % of the interfaces (Fig. 5b).'*

p 10, where are these example profiles from?

*We have added a paragraph with more details on the profiles.*

> *Lines 162-165:*
> *'In line with previous results (Rudels, 2015), staircases in the diffusive-convective regime (Fig. 7a) are mainly detected on the thermocline with the conservative temperature increasing with depth. These staircases are predominantly located in the Arctic Ocean at a depth between 300-400 m, which is between the warm and saline Atlantic Water and cold and fresh surface waters (Rudels, 2015).'*

*Lines 175-178:*
*'Thermohaline staircases with a high number of steps in the salt-finger regime are detected on the main thermocline where the conservative temperature decreases with depth (Fig. 7b). Compared to the staircases in the diffusive-convective regime, these staircases are located slightly deeper at 400-700 m. While the locations of these staircases vary, they are located above the cold and fresh Antarctic Intermediate Water, which is observed below 700 m (Tsuchiya, 1989; Fine, 1993; Talley, 1996).'*

p 11 "optimalization" should be "optimization"

*Corrected (line 213).*

p 13, line 219–220, I am not sure what is being meant here. It seems a lot of physical assumptions have been made that are implicit in the parameter choices.

*We meant that we, in contrast to previous detection algorithms, do not select on the Turner angle. We rephrased the sentence:*

> *Lines 257-258:*
> *'Note that by formulating the algorithm solely on this vertical structure of the staircases, we could use the Turner angle of the detected staircases for verification.'*

p 14, line 225 should say "both double-diffusive regimes" I believe Table A1, Julian should be capitalized and density should not be

*Corrected (line 270).*

Many of the references have incorrectly capitalized titles or journal names.

*Corrected.*

[Figure]

*Figure A 1 Steps of the detection algorithm applied on a profile in the Arctic Ocean, where steps are indicated on separate (a) conservative temperature and (b) absolute salinity profiles. Each profile is shifted for clarity. Similar to Figures 3-5, an interface is not considered by the detection algorithm when the interface characteristics did not meet the requirements of a previous step. Original profile is taken from Ice-Tethered-Profiler ITP64 at 137.8°W and 75.2°N on 29 January 2013. The details of the pre-processing and the algorithm steps are discussed in Section 2 and Section 3, respectively.*

[Figure]

*Figure A 2 as Figure A1, but for a profile in the Mediterranean Sea. Original profile is taken from Argo float 6901769 at 8.9ºE and 37.9ºN on 31 October 2017.*

[Figure]

*Figure A 3 as Figure A1, but for a profile in the western tropical North Atlantic. Original profile is taken from Argo float 4901478 at 53.3ºW and 11.6ºN on 9 August 2014.*

[Figure]

*Figure A1 Structure of the software. Each step in the software is shown by a box. Whenever a particular step is contained inside a function, the name of the function is mentioned above the step. Details of the preprocessing of the data and the detection algorithm are discussed in Sections 2 and Section 3, respectively.*